# Exploring the Space of Black-box Attacks on Deep Neural Networks

## Abstract

Existing black-box attacks on deep neural networks (DNNs) so far have largely focused on transferability, where an adversarial instance generated for a locally trained model can "transfer" to attack other learning models. In this paper, we propose novel Gradient Estimation black-box attacks for adversaries with query access to the target model's class probabilities, which do not rely on transferability. We also propose strategies to decouple the number of queries required to generate each adversarial sample from the dimensionality of the input. An iterative variant of our attack achieves close to 100% adversarial success rates for both targeted and untargeted attacks on DNNs. We carry out extensive experiments for a thorough comparative evaluation of black-box attacks and show that the proposed Gradient Estimation attacks outperform all transferability based black-box attacks we tested on both MNIST and CIFAR-10 datasets, achieving adversarial success rates similar to well known, state-of-the-art white-box attacks. We also apply the Gradient Estimation attacks successfully against a real-world content moderation classifier hosted by Clarifai. Furthermore, we evaluate black-box attacks against state-of-the-art defenses. We show that the Gradient Estimation attacks are very effective even against these defenses.

## 1 Introduction

The ubiquity of machine learning provides adversaries with both opportunities and incentives to develop strategic approaches to fool learning systems and achieve their malicious goals. Many attack strategies devised so far to generate adversarial examples to fool learning systems have been in the white-box setting, where adversaries are assumed to have access to the learning model (Szegedy et al. (2014); Goodfellow et al. (2015); Carlini & Wagner (2017); Moosavi-Dezfooli et al. (2015)). However, in many realistic settings, adversaries may only have black-box access to the model, i.e. they have no knowledge about the details of the learning system such as its parameters, but they may have query access to the model's predictions on input samples, including class probabilities. For example, we find this to be the case in some popular commercial AI offerings, such as those from IBM, Google and Clarifai. With access to query outputs such as class probabilities, the training loss of the target model can be found, but without access to the entire model, the adversary cannot access the gradients required to carry out white-box attacks.

Most existing black-box attacks on DNNs have focused on *transferability* based attacks (Papernot et al. (2016); Moosavi-Dezfooli et al. (2016); Papernot et al. (2017)), where adversarial examples crafted for a local surrogate model can be used to attack the target model to which the adversary has no direct access. The exploration of other black-box attack strategies is thus somewhat lacking so far in the literature. In this paper, we design powerful new black-box attacks using *limited query access to learning systems* which achieve adversarial success rates close to that of white-box attacks. These black-box attacks help us understand the extent of the threat posed to deployed systems by adversarial samples. The code to reproduce our results can be found at `https://github.com/anonymous`[1].

**New black-box attacks.** We propose novel *Gradient Estimation* attacks on DNNs, where the adversary is only assumed to have query access to the target model. These attacks do not need any

---

[1]Link anonymized for double-blind submission

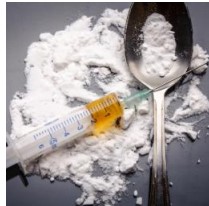 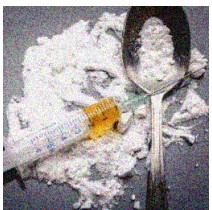

Figure 1: Sample adversarial images of Gradient Estimation attacks on Clarifai's Content Moderation model. **Left**: original image, classified as 'drug' with a confidence of 0.99. **Right**: adversarial sample with $\epsilon = 32$, classified as 'safe' with a confidence of 0.96.

access to a representative dataset or any knowledge of the target model architecture. In the Gradient Estimation attacks, the adversary adds perturbations proportional to the *estimated gradient*, instead of the true gradient as in white-box attacks (Goodfellow et al. (2015); Kurakin et al. (2016)). Since the direct Gradient Estimation attack requires a number of queries on the order of the dimension of the input, we explore strategies for reducing the number of queries to the target model. We also experimented with Simultaneous Perturbation Stochastic Approximation (SPSA) and Particle Swarm Optimization (PSO) as alternative methods to carry out query-based black-box attacks but found Gradient Estimation to work the best.

**Query-reduction strategies** We propose two strategies: *random feature grouping* and *principal component analysis (PCA) based query reduction*. In our experiments with the Gradient Estimation attacks on state-of-the-art models on MNIST (784 dimensions) and CIFAR-10 (3072 dimensions) datasets, we find that they match white-box attack performance, achieving attack success rates up to 90% for single-step attacks in the untargeted case and up to 100% for iterative attacks in both targeted and untargeted cases. We achieve this performance with just 200 to 800 queries per sample for single-step attacks and around 8,000 queries for iterative attacks. This is much fewer than the closest related attack by Chen et al. (2017). While they achieve similar success rates as our attack, the running time of their attack is up to $160\times$ longer for each adversarial sample (see Appendix I.6). A further advantage of the Gradient Estimation attack is that it does not require the adversary to train a local model, which could be an expensive and complex process for real-world datasets, in addition to the fact that training such a local model may require even more queries based on the training data.

**Attacking real-world systems**. To demonstrate the effectiveness of our Gradient Estimation attacks in the real world, we also carry out a practical black-box attack using these methods against the Not Safe For Work (NSFW) classification and Content Moderation models developed by Clarifai, which we choose due to their socially relevant application. These models have begun to be deployed for real-world moderation (Liu, 2016), which makes such black-box attacks especially pernicious. We carry out these attacks with *no knowledge of the training set*. We have demonstrated successful attacks (Figure 1) with just around 200 queries per image, taking around a minute per image. In Figure 1, the target model classifies the adversarial image as 'safe' with high confidence, in spite of the content that had to be moderated still being clearly visible. We note here that *due to the nature of the images we experiment with, we only show one example here*, as the others may be offensive to readers. The full set of images is hosted anonymously at `https://www.dropbox.com/s/xsu31tjr0yq7rj7/clarifai-examples.zip?dl=0`.

**Comparative evaluation of black-box attacks.** We carry out a thorough empirical comparison of various black-box attacks (given in Table 7) on both MNIST and CIFAR-10 datasets. We study attacks that require zero queries to the learning model, including the addition of perturbations that are either random or proportional to the difference of means of the original and targeted classes, as well as various transferability based black-box attacks. We show that the proposed Gradient Estimation attacks outperform other black-box attacks in terms of attack success rate and achieve results comparable with white-box attacks.

In addition, we also evaluate the effectiveness of these attacks on DNNs made more robust using adversarial training (Goodfellow et al., 2015; Szegedy et al., 2014) and its recent variants including ensemble adversarial training (Tramèr et al., 2017a) and iterative adversarial training (Mądry et al., 2017). We find that although standard and ensemble adversarial training confer some robustness against single-step attacks, they are vulnerable to iterative Gradient Estimation attacks, with adversar-

ial success rates in excess of 70% for both targeted and untargeted attacks. We find that our methods outperform other black-box attacks and achieve performance comparable to white-box attacks.

**Related Work.** Existing black-box attacks that do not use a local model were first proposed for convex inducing two-class classifiers by Nelson et al. (2012). For malware data, Xu et al. (2016) use genetic algorithms to craft adversarial samples, while Dang et al. (2017) use hill climbing algorithms. These methods are prohibitively expensive for non-categorical and high-dimensional data such as images. Papernot et al. (2017) proposed using queries to a target model to train a local surrogate model, which was then used to to generate adversarial samples. This attack relies on transferability. To the best of our knowledge, the only previous literature on query-based black-box attacks in the deep learning setting is independent work by Narodytska & Kasiviswanathan (2016) and Chen et al. (2017).

Narodytska & Kasiviswanathan (2016) propose a greedy local search to generate adversarial samples by perturbing randomly chosen pixels and using those which have a large impact on the output probabilities. Their method uses 500 queries per iteration, and the greedy local search is run for around 150 iterations for each image, resulting in a total of 75,000 queries per image, which is much higher than any of our attacks. Further, we find that our methods achieve higher targeted and untargeted attack success rates on both MNIST and CIFAR-10 as compared to their method. Chen et al. (2017) propose a black-box attack method named ZOO, which also uses the method of finite differences to estimate the derivative of a function. However, while we propose attacks that compute an adversarial perturbation, approximating FGSM and iterative FGS; ZOO approximates the Adam optimizer, while trying to perform coordinate descent on the loss function proposed by Carlini & Wagner (2017). Neither of these works demonstrates the effectiveness of their attacks on real-world systems or on state-of-the-art defenses.

## 2 BACKGROUND AND EVALUATION SETUP

In this section, we will first introduce the notation we use throughout the paper and then describe the evaluation setup and metrics used in the remainder of the paper.

### 2.1 NOTATION

A classifier $f(\cdot; \theta) : \mathcal{X} \to \mathcal{Y}$ is a function mapping from the domain $\mathcal{X}$ to the set of classification outputs $\mathcal{Y}$. ($\mathcal{Y} = \{0, 1\}$ in the case of binary classification, i.e. $\mathcal{Y}$ is the set of class labels.) The number of possible classification outputs is then $|\mathcal{Y}|$. $\theta$ is the set of parameters associated with a classifier. Throughout, the target classifier is denoted as $f(\cdot; \theta)$, but the dependence on $\theta$ is dropped if it is clear from the context. $\mathcal{H}$ denotes the constraint set which an adversarial sample must satisfy. $\ell_f(\mathbf{x}, y)$ is used to represent the loss function for the classifier $f$ with respect to inputs $\mathbf{x} \in \mathcal{X}$ and their true labels $y \in \mathcal{Y}$.

Since the black-box attacks we analyze focus on neural networks in particular, we also define some notation specifically for neural networks. The outputs of the penultimate layer of a neural network $f$, representing the output of the network computed sequentially over all preceding layers, are known as the logits. We represent the logits as a vector $\phi^f(\mathbf{x}) \in \mathbb{R}^{|\mathcal{Y}|}$. The final layer of a neural network $f$ used for classification is usually a softmax layer represented as a vector of probabilities $\mathbf{p}^f(\mathbf{x}) = [p_1^f(\mathbf{x}), \dots, p_{|\mathcal{Y}|}^f(\mathbf{x})]$, with $\sum_{i=1}^{|\mathcal{Y}|} p_i^f(\mathbf{x}) = 1$ and $p_i^f(\mathbf{x}) = \frac{e^{\phi_i^f(\mathbf{x})}}{\sum_{j=1}^{|\mathcal{Y}|} e^{\phi_j^f(\mathbf{x})}}$.

### 2.2 EVALUATION SETUP FOR MNIST AND CIFAR-10

The empirical evaluation carried out in Section 3 is on state-of-the-art neural networks on the MNIST (LeCun & Cortes, 1998) and CIFAR-10 (Krizhevsky & Hinton, 2009) datasets. The details of the datasets are given in Appendix C.1, and the architecture and training details for all models are given in Appendix C.2. Only results for untargeted attacks are given in the main body of the paper. All results for targeted attacks are contained in Appendix E. We use two different loss functions in our evaluation, the standard cross-entropy loss (abbreviated as xent) and the logit-based loss (ref. Section 3.1.2, abbreviated as logit). In all of these attacks, the adversary's perturbation is constrained using the $L_\infty$ distance.

The details of baseline black-box attacks and results can be found in Appendix A.1.1. Similarly, detailed descriptions and results for transferability-based attacks are in Appendix A.2. The full set of attacks that was evaluated is given in Table 7 in Appendix G, which also provides a taxonomy for black-box attacks.

**MNIST.** Each pixel of the MNIST image data is scaled to $[0, 1]$. We trained four different models on the MNIST dataset, denoted Models A to D, which are used by Tramèr et al. (2017a) and represent a good variety of architectures. For the attacks constrained with the $L_\infty$ distance, we vary the adversary's perturbation budget $\epsilon$ from 0 to 0.4, since at a perturbation budget of 0.5, any image can be made solid gray.

**CIFAR-10.** Each pixel of the CIFAR-10 image data is in $[0, 255]$. We choose three model architectures for this dataset, which we denote as Resnet-32, Resnet-28-10 (ResNet variants (He et al., 2016; Zagoruyko & Komodakis, 2016)), and Std.-CNN (a standard CNN[2] from Tensorflow (Abadi et al., 2015)). For the attacks constrained with the $L_\infty$ distance, we vary the adversary's perturbation budget $\epsilon$ from 0 to 28.

## 2.3 METRICS

Throughout the paper, we use standard metrics to characterize the effectiveness of various attack strategies. For *MNIST*, all metrics for single-step attacks are computed with respect to the *test set* consisting of 10,000 samples, while metrics for iterative attacks are computed with respect to the first 1,000 samples from the test set. For the *CIFAR-10* data, we choose 1,000 random samples from the test set for single-step attacks and a 100 random samples for iterative attacks. In our evaluations of targeted attacks, we choose target $T$ for each sample uniformly at random from the set of classification outputs, except the true class $y$ of that sample.

**Attack success rate.** The main metric, the attack success rate, is the fraction of samples that meets the adversary's goal: $f(\mathbf{x}_{\text{adv}}) \neq y$ for untargeted attacks and $f(\mathbf{x}_{\text{adv}}) = T$ for targeted attacks with target $T$ (Szegedy et al., 2014; Tramèr et al., 2017a). Alternative evaluation metrics are discussed in Appendix C.3.

**Average distortion.** We also evaluate the average distortion for adversarial examples using average $L_2$ distance between the benign samples and the adversarial ones as suggested by Gu & Rigazio (2014): $\Delta(\mathbf{X}_{\text{adv}}, \mathbf{X}) = \frac{1}{N} \sum_{i=1}^{N} \|(\mathbf{X}_{\text{adv}})_i - (\mathbf{X})_i\|_2$ where $N$ is the number of samples. This metric allows us to compare the average distortion for attacks which achieve similar attack success rates, and therefore infer which one is stealthier.

**Number of queries.** Query based black-box attacks make queries to the target model, and this metric may affect the cost of mounting the attack. This is an important consideration when attacking real-world systems which have costs associated with the number of queries made.

## 3 QUERY BASED ATTACKS: GRADIENT ESTIMATION ATTACK

Deployed learning systems often provide feedback for input samples provided by the user. Given query feedback, different adaptive, query-based algorithms can be applied by adversaries to understand the system and iteratively generate effective adversarial examples to attack it. Formal definitions of query-based attacks are in Appendix D. We initially explored a number of methods of using query feedback to carry out black-box attacks including Particle Swarm Optimization (Kennedy, 2011) and Simultaneous Perturbation Stochastic Approximation (Spall, 1992). However, these methods were not effective at finding adversarial examples for reasons detailed in Section 3.4, which also contains the results obtained.

Given the fact that many white-box attacks for generating adversarial examples are based on gradient information, we then tried *directly estimating the gradient to carry out black-box attacks*, and found it to be very effective in a range of conditions. In other words, the adversary can approximate white-box Single-step and Iterative FGSM attacks (Goodfellow et al., 2015; Kurakin et al., 2016) using estimates of the losses that are needed to carry out those attacks. We first propose a Gradient

---

[2]https://github.com/tensorflow/models/tree/master/tutorials/image/cifar10

Estimation black-box attack based on the method of finite differences (Spall, 2005). The drawback of a naive implementation of the finite difference method, however, is that it requires $O(d)$ queries per input, where $d$ is the dimension of the input. This leads us to explore methods such as random grouping of features and feature combination using components obtained from Principal Component Analysis (PCA) to reduce the number of queries.

**Threat model and justification.** We assume that the adversary can obtain the vector of output probabilities for any input $\mathbf{x}$. The set of queries the adversary can make is then $\mathcal{Q}_f = \{\mathbf{p}^f(\mathbf{x}), \forall \mathbf{x}\}$. Note that an adversary with access to the softmax probabilities will be able to recover the logits up to an additive constant, by taking the logarithm of the softmax probabilities. For untargeted attacks, the adversary only needs access to the output probabilities for the two most likely classes.

A compelling reason for assuming this threat model for the adversary is that many existing cloud-based ML services allow users to query trained models (Watson Visual Recognition, Clarifai, Google Vision API). The results of these queries are confidence scores which can be used to carry out Gradient Estimation attacks. These trained models are often deployed by the clients of these ML as a service (MLaaS) providers (Liu (2016)). Thus, an adversary can pose as a user for a MLaaS provider and create adversarial examples using our attack, which can then be used against any client of that provider.

## 3.1 FINITE DIFFERENCE METHOD FOR GRADIENT ESTIMATION

In this section, we focus on the method of finite differences to carry out Gradient Estimation based attacks. All the analysis and results are presented for untargeted attacks, but can be easily extended to targeted attacks (Appendix E). Let the function whose gradient is being estimated be $g(\mathbf{x})$. The input to the function is a $d$-dimensional vector $\mathbf{x}$, whose elements are represented as $\mathbf{x}_i$, where $i \in [1, \ldots, d]$. The canonical basis vectors are represented as $\mathbf{e}_i$, where $\mathbf{e}_i$ is 1 only in the $i^{th}$ component and 0 everywhere else. Then, a two-sided estimation of the gradient of $g$ with respect to $\mathbf{x}$ is given by

$$\text{FD}_{\mathbf{x}}(g(\mathbf{x}), \delta) = \begin{bmatrix} \frac{g(\mathbf{x}+\delta\mathbf{e}_1)-g(\mathbf{x}-\delta\mathbf{e}_1)}{2\delta} \\ \vdots \\ \frac{g(\mathbf{x}+\delta\mathbf{e}_d)-g(\mathbf{x}-\delta\mathbf{e}_d)}{2\delta} \end{bmatrix}. \tag{1}$$

$\delta$ is a free parameter that controls the accuracy of the estimation. A one-sided approximation can also be used, but will be less accurate (Wright & Nocedal, 1999). If the gradient of the function $g$ exists, then $\lim_{\delta \to 0} \text{FD}_{\mathbf{x}}(g(\mathbf{x}), \delta) = \nabla_{\mathbf{x}} g(\mathbf{x})$. The finite difference method is useful for a black-box adversary aiming to approximate a gradient based attack, since the gradient can be directly estimated with access to only the function values.

### 3.1.1 APPROXIMATE FGS WITH FINITE DIFFERENCES

In the untargeted FGS method, the gradient is usually taken with respect to the cross-entropy loss between the true label of the input and the softmax probability vector. The cross-entropy loss of a network $f$ at an input $\mathbf{x}$ is then $\ell_f(\mathbf{x}, y) = -\sum_{j=1}^{|\mathcal{Y}|} \mathbf{1}[j = y] \log p_j^f(\mathbf{x}) = -\log p_y^f(\mathbf{x})$, where $y$ is the index of the original class of the input. The gradient of $\ell_f(\mathbf{x}, y)$ is

$$\nabla_{\mathbf{x}} \ell_f(\mathbf{x}, y) = -\frac{\nabla_{\mathbf{x}} p_y^f(\mathbf{x})}{p_y^f(\mathbf{x})}. \tag{2}$$

An adversary with query access to the softmax probabilities then just has to estimate the gradient of $p_y^f(\mathbf{x})$ and plug it into Eq. 2 to get the estimated gradient of the loss. The adversarial sample thus generated is

$$\mathbf{x}_{\text{adv}} = \mathbf{x} + \epsilon \cdot \text{sign}\left(\frac{\text{FD}_{\mathbf{x}}(p_y^f(\mathbf{x}), \delta)}{p_y^f(\mathbf{x})}\right). \tag{3}$$

This method of generating adversarial samples is denoted as FD-xent.

### 3.1.2 ESTIMATING THE LOGIT-BASED LOSS

We also use a loss function based on logits which was found to work well for white-box attacks by Carlini & Wagner (2017). The loss function is given by

$$\ell(\mathbf{x}, y) = \max(\phi(\mathbf{x} + \boldsymbol{\delta})_y - \max\{\phi(\mathbf{x} + \boldsymbol{\delta})_i : i \neq y\}, -\kappa), \quad (4)$$

where $y$ represents the ground truth label for the benign sample $\mathbf{x}$ and $\phi(\cdot)$ are the logits. $\kappa$ is a confidence parameter that can be adjusted to control the strength of the adversarial perturbation. If the confidence parameter $\kappa$ is set to 0, the logit loss is $\max(\phi(\mathbf{x} + \boldsymbol{\delta})_y - \max\{\phi(\mathbf{x} + \boldsymbol{\delta})_i : i \neq y\}, 0)$. For an input that is correctly classified, the first term is always greater than 0, and for an incorrectly classified input, an untargeted attack is not meaningful to carry out. Thus, the loss term reduces to $\phi(\mathbf{x} + \boldsymbol{\delta})_y - \max\{\phi(\mathbf{x} + \boldsymbol{\delta})_i : i \neq y\}$ for relevant inputs.

An adversary can compute the logit values up to an additive constant by taking the logarithm of the softmax probabilities, which are assumed to be available in this threat model. Since the loss function is equal to the difference of logits, the additive constant is canceled out. Then, the finite differences method can be used to estimate the difference between the logit values for the original class $y$, and the second most likely class $y'$, i.e., the one given by $y' = \operatorname{argmax}_{i \neq y} \phi(\mathbf{x})_i$. The untargeted adversarial sample generated for this loss in the white-box case is $\mathbf{x}_{\mathrm{adv}} = \mathbf{x} + \epsilon \cdot \operatorname{sign}(\nabla_{\mathbf{x}}(\phi(\mathbf{x})_{y'} - \phi(\mathbf{x})_y))$. Similarly, in the case of a black-box adversary with query-access to the softmax probabilities, the adversarial sample is

$$\mathbf{x}_{\mathrm{adv}} = \mathbf{x} + \epsilon \cdot \operatorname{sign}(\mathrm{FD}_{\mathbf{x}}(\phi(\mathbf{x})_{y'} - \phi(\mathbf{x})_y, \delta)). \quad (5)$$

This attack is denoted as FD-logit.

| MNIST | Baseline | | Gradient Estimation using Finite Differences | | | | Transfer from Model B | | | |
|---|---|---|---|---|---|---|---|---|---|---|
| | | | Single-step | | Iterative | | Single-step | | Iterative | |
| Model | D. of M. | Rand. | FD-xent | FD-logit | IFD-xent | IFD-logit | FGS-xent | FGS-logit | IFGS-xent | IFGS-logit |
| A | 44.8 (5.6) | 8.5 (6.1) | 51.6 (3.3) | 92.9 (6.1) | 75.0 (3.6) | **100.0** (2.1) | 66.3 (6.2) | 80.8 (6.3) | 89.8 (4.75) | 88.5 (4.75) |
| B | 81.5 (5.6) | 7.8 (6.1) | 69.2 (4.5) | 98.9 (6.3) | 86.7 (3.9) | **100.0** (1.6) | - | - | - | - |
| C | 20.2 (5.6) | 4.1 (6.1) | 60.5 (3.8) | 86.1 (6.2) | 80.2 (4.5) | **100.0** (2.2) | 49.5 (6.2) | 57.0 (6.3) | 79.5 (4.75) | 78.7 (4.75) |
| D | 97.1 (5.6) | 38.5 (6.1) | 95.4 (5.8) | **100.0** (6.1) | 98.4 (5.4) | **100.0** (1.2) | 76.3 (6.2) | 87.6 (6.3) | 73.3 (4.75) | 71.4 (4.75) |
| **CIFAR-10** | **Baseline** | | **Gradient Estimation using Finite Differences** | | | | **Transfer from Resnet-28-10** | | | |
| | | | Single-step | | Iterative | | Single-step | | Iterative | |
| Model | D. of M. | Rand. | FD-xent | FD-logit | IFD-xent | IFD-logit | FGS-xent | FGS-logit | IFGS-xent | IFGS-logit |
| Resnet-32 | 9.3 (440.5) | 19.4 (439.4) | 49.1 (217.1) | 86.0 (410.3) | 62.0 (149.9) | **100.0** (65.7) | 74.5 (439.4) | 76.6 (439.4) | 99.0 (275.4) | 98.9 (275.6) |
| Resnet-28-10 | 6.7 (440.5) | 17.1 (439.4) | 50.1 (214.8) | 88.2 (421.6) | 46.0 (120.4) | **100.0** (74.9) | - | - | - | - |
| Std.-CNN | 20.3 (440.5) | 22.2 (439.4) | 80.0 (341.3) | 98.9 (360.9) | 66.0 (202.5) | **100.0** (79.9) | 37.4 (439.4) | 37.7 (439.4) | 33.7 (275.4) | 33.6 (275.6) |

Table 1: **Untargeted black-box attacks**: Each entry has the attack success rate for the attack method given in that column on the model in each row. The number in parentheses for each entry is $\Delta(\mathbf{X}, \mathbf{X}_{\mathrm{adv}})$, the average distortion over all samples used in the attack. In each row, the entry in **bold** represents the black-box attack with the best performance on that model. Gradient Estimation using Finite Differences is our method, which has performance matching white-box attacks. **Above**: MNIST, $L_\infty$ constraint of $\epsilon = 0.3$. **Below**: CIFAR-10, $L_\infty$ constraint of $\epsilon = 8$.

### 3.1.3 ITERATIVE ATTACKS WITH ESTIMATED GRADIENTS

The iterative variant of the gradient based attack described in Section A.1.2 is a powerful attack that often achieves much higher attack success rates in the white-box setting than the simple single-step gradient based attacks. Thus, it stands to reason that a version of the iterative attack with estimated gradients will also perform better than the single-step attacks described until now. An iterative attack with $t + 1$ iterations using the cross-entropy loss is:

$$\mathbf{x}_{\mathrm{adv}}^{t+1} = \Pi_{\mathcal{H}} \left( \mathbf{x}_{\mathrm{adv}}^t + \alpha \cdot \operatorname{sign} \left( \frac{\mathrm{FD}_{\mathbf{x}_{\mathrm{adv}}^t} p_y^f(\mathbf{x}_{\mathrm{adv}}^t)}{p_y^f(\mathbf{x}_{\mathrm{adv}}^t)} \right) \right), \quad (6)$$

where $\alpha$ is the step size and $\mathcal{H}$ is the constraint set for the adversarial sample. This attack is denoted as IFD-xent. If the logit loss is used instead, it is denoted as IFD-logit.

### 3.1.4 EVALUATION OF GRADIENT ESTIMATION USING FINITE DIFFERENCES

In this section, we summarize the results obtained using Gradient Estimation attacks with Finite Differences and describe the parameter choices made.

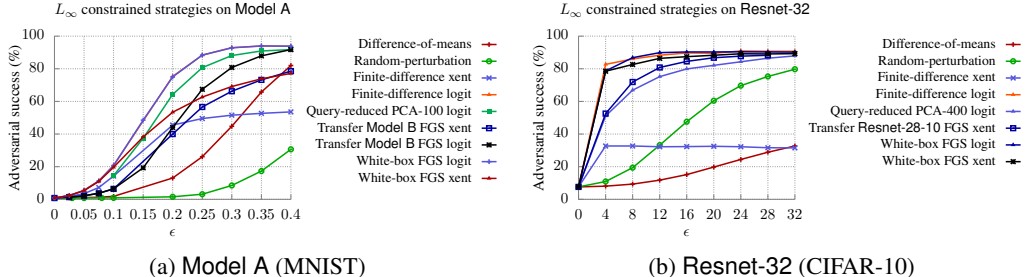

Figure 2: **Effectiveness of various single step black-box attacks on Model A (MNIST) and Resnet-32 (CIFAR-10)**. The y-axis for both figures gives the variation in adversarial success as $\epsilon$ is increased. The most successful black-box attack strategy in both cases is the Gradient Estimation attack using Finite Differences with the logit loss (FD-logit), which coincides almost exactly with the white-box FGS attack with the logit loss (WB FGS-logit). Also, the Gradient Estimation attack with query reduction using PCA (GE-QR (PCA-$k$, logit)) performs well for both datasets as well.

**FD-logit and IFD-logit match white-box attack adversarial success rates**: The Gradient Estimation attack with Finite Differences (FD-logit) is the most successful *untargeted* single-step black-box attack for MNIST and CIFAR-10 models. It significantly outperforms transferability-based attacks (Table 1) and closely tracks white-box FGS with a logit loss (WB FGS-logit) on MNIST and CIFAR-10 (Figure 2). For adversarial samples generated iteratively, the Iterative Gradient Estimation attack with Finite Differences (IFD-logit) achieves 100% adversarial success rate across all models on both datasets (Table 1). We used 0.3 for the value of $\epsilon$ for the MNIST dataset and 8 for the CIFAR-10 dataset. The average distortion for both FD-logit and IFD-logit closely matches their white-box counterparts, FGS-logit and IFGS-logit as given in Table 8.

**FD-T and IFD-T achieve the highest adversarial success rates in the targeted setting**: For *targeted* black-box attacks, IFD-xent-T achieves 100% adversarial success rates on almost all models as shown by the results in Table 6. While FD-xent-T only achieves about 30% adversarial success rates, this matches the performance of single-step white-box attacks such as FGS-xent-T and FGS-logit-T (Table 9). The average distortion for samples generated using gradient estimation methods is similar with that of white-box attacks.

**Parameter choices**: We use $\delta = 1.0$ for FD-xent and IFD-xent for both datasets, while using $\delta = 0.01$ for FD-logit and IFD-logit. We find that a larger value of $\delta$ is needed for xent loss based attacks to work. The reason for this is that the probability values used in the xent loss are not as sensitive to changes as in the logit loss, and thus the gradient cannot be estimated since the function value does not change at all when a single pixel is perturbed. For the Iterative Gradient Estimation attacks using Finite Differences, we use $\alpha = 0.01$ and $t = 40$ for the MNIST results and $\alpha = 1.0$ and $t = 10$ for CIFAR-10 throughout. The same parameters are used for the white-box Iterative FGS attack results given in Appendix I.1. This translates to 62720 queries for MNIST (40 steps of iteration) and 61440 queries (10 steps of iteration) for CIFAR-10 per sample. We find these choices work well, and keep the running time of the Gradient Estimation attacks at a manageable level. However, we find that we can achieve similar adversarial success rates with much fewer queries using query reduction methods which we describe in the next section.

## 3.2 QUERY REDUCTION

The major drawback of the approximation based black-box attacks is that the number of queries needed per adversarial sample is large. For an input with dimension $d$, the number of queries will be exactly $2d$ for a two-sided approximation. This may be too large when the input is high-dimensional. So we examine two techniques in order to reduce the number of queries the adversary has to make. Both techniques involve estimating the gradient for groups of features, instead of estimating it one feature at a time.

The justification for the use of feature grouping comes from the relation between gradients and directional derivatives (Hildebrand, 1962) for differentiable functions. The directional derivative of a function $g$ is defined as $\nabla_{\mathbf{v}} g(\mathbf{x}) = \lim_{h \to 0} \frac{g(\mathbf{x}+h\mathbf{v})-g(\mathbf{x})}{h}$. It is a generalization of a partial derivative. For differentiable functions, $\nabla_{\mathbf{v}} g(\mathbf{x}) = \nabla_{\mathbf{x}} g(\mathbf{x}) \cdot \mathbf{v}$, which implies that the directional derivative is just the projection of the gradient along the direction $\mathbf{v}$. Thus, estimating the gradient by grouping features is equivalent to estimating an approximation of the gradient constructed by projecting it along appropriately chosen directions. The estimated gradient $\hat{\nabla}_{\mathbf{x}} g(\mathbf{x})$ of any function $g$ can be computed using the techniques below, and then plugged in to Equations 3 and 5 instead of the finite difference term to create an adversarial sample. Next, we introduce the techniques applied to group the features for estimation. Detailed algorithms for these techniques are given in Appendix F.

### 3.2.1 QUERY REDUCTION BASED ON RANDOM GROUPING

The simplest way to group features is to choose, without replacement, a random set of features. The gradient can then be simultaneously estimated for all these features. If the size of the set chosen is $k$, then the number of queries the adversary has to make is $\lceil \frac{d}{k} \rceil$. When $k = 1$, this reduces to the case where the partial derivative with respect to every feature is found, as in Section 3.1. In each iteration of Algorithm 1, there is a set of indices $S$ according to which $\mathbf{v}$ is determined, with $\mathbf{v}_i = 1$ if and only if $i \in S$. Thus, the directional derivative being estimated is $\sum_{i \in S} \frac{\partial g(\mathbf{x})}{\partial \mathbf{x}_i}$, which is an average of partial derivatives. Thus, the quantity being estimated is not the gradient itself, but an index-wise averaged version of it.

### 3.2.2 QUERY REDUCTION USING PCA COMPONENTS

A more principled way to reduce the number of queries the adversary has to make to estimate the gradient is to compute directional derivatives along the principal components as determined by principal component analysis (PCA) (Shlens, 2014), which requires the adversary to have access to a set of data which is represetative of the training data. A more detailed description of PCA and the Gradient Estimation attack using PCA components for query reduction is given in Appendix F.2. In Algorithm 2, $\mathbf{U}$ is the $d \times d$ matrix whose columns are the principal components $\mathbf{u}_i$, where $i \in [d]$. The quantity being estimated in Algorithm 2 in the Appendix is an approximation of the gradient in the PCA basis:

$$(\nabla_{\mathbf{x}} g(\mathbf{x}))^k = \sum_{i=1}^{k} \left( \nabla_{\mathbf{x}} g(\mathbf{x})^{\mathsf{T}} \frac{\mathbf{u}_i}{\|\mathbf{u}_i\|} \right) \frac{\mathbf{u}_i}{\|\mathbf{u}_i\|},$$

where the term on the left represents an approximation of the true gradient by the sum of its projection along the top $k$ principal components. In Algorithm 2, the weights of the representation in the PCA basis are approximated using the approximate directional derivatives along the principal components.

## 3.3 ITERATIVE ATTACKS WITH QUERY REDUCTION

Performing an iterative attack with the gradient estimated using the finite difference method (Equation 1) could be expensive for an adversary, needing $2td$ queries to the target model, for $t$ iterations with the two-sided finite difference estimation of the gradient. To lower the number of queries needed, the adversary can use either of the query reduction techniques described above to reduce the number of queries to $2tk$ ( $k < d$ ). These attacks using the cross-entropy loss are denoted as IGE-QR (RG-$k$, xent) for the random grouping technique and IGE-QR (PCA-$k$, xent) for the PCA-based technique.

### 3.3.1 EVALUATION OF GRADIENT ESTIMATION ATTACKS WITH QUERY REDUCTION

In this section, we summarize the results obtained using Gradient Estimation attacks with query reduction.

**Gradient estimation with query reduction maintains high attack success rates**: For both datasets, the Gradient Estimation attack with PCA based query reduction (GE-QR (PCA-$k$, logit)) is effective, with performance close to that of FD-logit with $k = 100$ for MNIST (Figure 2a) and $k = 400$ for CIFAR-10 (Figure 2b). The Iterative Gradient Estimation attacks with both Random Grouping and PCA based query reduction (IGE-QR (RG-$k$, logit) and IGE-QR (PCA-$k$, logit)) achieve close to 100% success rates for untargeted attacks and above 80% for targeted attacks on Model A on MNIST

| Query-based attack | Attack success | No. of queries | Time per sample (s) |
|---|---|---|---|
| Finite Diff. | 92.9 (6.1) | 1568 | $8.8 \times 10^{-2}$ |
| Gradient Estimation (RG-8) | 61.5 (6.0) | 196 | $1.1 \times 10^{-2}$ |
| Iter. Finite Diff. | 100.0 (2.1) | 62720 | 3.5 |
| Iter. Gradient Estimation (RG-8) | 98.4 (1.9) | 8000 | 0.43 |
| Particle Swarm Optimization | 84.1 (5.3) | 10000 | 21.2 |
| SPSA | 96.7 (3.9) | 8000 | 1.25 |

Table 2: Comparison of **untargeted query-based black-box attack** methods. All results are for attacks using the first 1000 samples from the MNIST dataset on Model A and with an $L_\infty$ constraint of 0.3. The logit loss is used for all methods expect PSO, which uses the class probabilities.

and Resnet-32 on CIFAR-10 (Figure 3). Figure 3 clearly shows the effectiveness of the gradient estimation attack across models, datasets, and adversarial goals. While random grouping is not as effective as the PCA based method for Single-step attacks, it is as effective for iterative attacks. Thus, powerful black-box attacks can be carried out purely using query access.

### 3.4 OTHER QUERY-BASED ATTACKS

We experimented with Particle Swarm Optimization (PSO),[3] a commonly used evolutionary optimization strategy, to construct adversarial samples as was done by Sharif et al. (2016), but found it to be prohibitively slow for a large dataset, and it was unable to achieve high adversarial success rates even on the MNIST dataset. We also tried to use the Simultaneous Perturbation Stochastic Approximation (SPSA) method, which is similar to the method of Finite Differences, but it estimates the gradient of the loss along a *random direction* **r** at each step, instead of along the canonical basis vectors. While each step of SPSA only requires 2 queries to the target model, a large number of steps are nevertheless required to generate adversarial samples. A single step of SPSA does not reliably produce adversarial samples. The two main disadvantages of this method are that i) the convergence of SPSA is much more sensitive in practice to the choice of both $\delta$ (gradient estimation step size) and $\alpha$ (loss minimization step size), and ii) even with the same number of queries as the Gradient Estimation attacks, the attack success rate is lower even though the distortion is higher.

A comparative evaluation of all the query-based black-box attacks we experimented with for the MNIST dataset is given in Table 2. The PSO based attack uses class probabilities to define the loss function, as it was found to work better than the logit loss in our experiments. The attack that achieves the best trade-off between speed and attack success is IGE-QR (RG-$k$, logit).

Detailed evaluation results are contained in Appendix I. In particular, discussions of the results on baseline attacks (Appendix I.2), effect of dimension on query reduced Gradient Estimation attacks (Appendix I.4), Single-step attacks on defenses (Appendix I.5), and the efficiency of Gradient Estimation attacks (Appendix I.6) are provided. Sample adversarial examples are shown in Appendix H.

## 4 ATTACKING DEFENSES

In this section, we evaluate black-box attacks against different defenses based on adversarial training and its variants. Details about the adversarially trained models can be found in Appendix B. We focus on adversarial training based defenses as they aim to directly improve the robustness of DNNs, and are among the most effective defenses demonstrated so far in the literature. We also conduct real-world attacks on models deployed by Clarifai, a MlaaS provider.

In the discussion of our results, we focus on the attack success rate obtained by Iterative Gradient Estimation attacks, since they perform much better than any single-step black-box attack. Nevertheless, in Figure 6 and Appendix I.5, we show that with the addition of an initial random perturbation to overcome "gradient masking" (Tramèr et al., 2017a), the Gradient Estimation attack with Finite Differences is the most effective single-step black-box attack on adversarially trained models on MNIST.

---

[3]Using freely available code from `http://pythonhosted.org/pyswarm/`

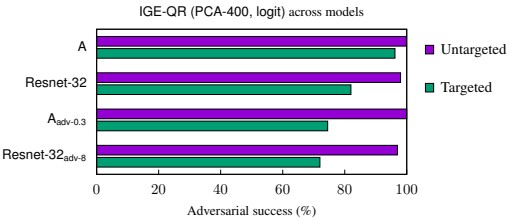

| MNIST | Single-step | | Iterative | |
|---|---|---|---|---|
| | PCA-100 | RG-8 | PCA-100 | RG-8 |
| A | 88.1 (6.0) | 61.5 (6.0) | 99.9 (2.5) | 98.4 (1.9) |
| A$_{adv-0.3}$ | 4.1 (5.8) | 2.0 (5.3) | 50.7 (4.2) | 27.5 (2.4) |
| **CIFAR-10** | Single-step | | Iterative | |
| | PCA-400 | RG-8 | PCA-400 | RG-8 |
| Resnet-32 | 66.9 (410.7) | 66.8 (402.7) | 98.0 (140.7) | 99.0 (80.5) |
| Resnet-32 $_{adv-8}$ | 8.0 (402.1) | 7.7 (401.8) | 97.0 (151.3) | 98.0 (92.9) |

(a) Adversarial success rates for **untargeted attacks using query reduction**. The parameters used for query reduction are indicated in the table.

(b) Adversarial success rates of the **Iterative Gradient Estimation attack using the logit loss, with PCA (400) used for query reduction**.

Figure 3: **Adversarial success rates for query-reduced attacks**. $\epsilon$ is set to 0.3 for Model A on MNIST and 8 for Resnet-32 on CIFAR-10. Model A$_{adv-0.3}$ and Resnet-32 $_{adv-8}$ are adversarially trained variants (ref. Section B) of the models. All attacks use roughly 8000 queries per sample for both datasets.

## 4.1 MNIST SETUP AND RESULTS

We train variants of Model A with the 3 adversarial training strategies described in Appendix B using adversarial samples based on an $L_\infty$ constraint of 0.3. Model A$_{adv-0.3}$ is trained with FGS samples, while Model A$_{adv-iter-0.3}$ is trained with iterative FGS samples using $t = 40$ and $\alpha = 0.01$. For the model with ensemble training, Model A$_{adv-ens-0.3}$ is trained with pre-generated FGS samples for Models A, C, and D, as well as FGS samples. The source of the samples is chosen randomly for each minibatch during training.

**Evaluation of iterative attacks on different adversarial training defenses**: While single-step black-box attacks are less effective at $\epsilon$ lower than the one used for training, our experiments show that iterative black-box attacks continue to work well even against adversarially trained networks. For example, the Iterative Gradient Estimation attack using Finite Differences with a logit loss (IFD-logit) achieves an adversarial success rate of 96.4% against Model A$_{adv-ens-0.3}$, while the best transferability attack has a success rate of 4.9%. It is comparable to the white-box attack success rate of 93% from Table 10. However, Model A$_{adv-iter-0.3}$ is quite robust even against iterative attacks, with the highest black-box attack success rate achieved being 14.5%.

Further, in Figure 3, we can see that using just 4000 queries per sample, the Iterative Gradient Estimation attack using PCA for query reduction (IGE-QR (PCA-400, logit)) achieves 100% (untargeted) and 74.5% (targeted) adversarial success rates against Model A$_{adv-0.3}$. Our methods far outperform the other black-box attacks, as shown in Table 10.

## 4.2 CIFAR-10 SETUP AND RESULTS

We train variants of Resnet-32 using adversarial samples with an $L_\infty$ constraint of 8. Resnet-32 $_{adv-8}$ is trained with FGS samples with the same constraint, and Resnet-32 $_{ens-adv-8}$ is trained with pre-generated FGS samples from Resnet-32 and Std.-CNN as well as FGS samples. Resnet-32 $_{adv-iter-8}$ is trained with iterative FGS samples using $t = 10$ and $\alpha = 1.0$.

Iterative black-box attacks perform well against adversarially trained models for CIFAR-10 as well. IFD-logit achieves attack success rates of 100% against both Resnet-32 $_{adv-8}$ and Resnet-32 $_{adv-ens-8}$ (Table 3), which reduces slightly to 97% when IFD-QR (PCA-400, logit) is used. This matches the performance of white-box attacks as given in Table 10. IFD-QR (PCA-400, logit) also achieves a 72% success rate for targeted attacks at $\epsilon = 8$ as shown in Figure 3.

The iteratively trained model has poor performance on both benign as well as adversarial samples. Resnet-32 $_{adv-iter-8}$ has an accuracy of only 79.1% on benign data, as shown in Table 4. The Iterative Gradient Estimation attack using Finite Differences with cross-entropy loss (IFD-xent) achieves an untargeted attack success rate of 55% on this model, which is lower than on the other adversarially trained models, but still significant. This is in line with the observation by Mądry et al. (2017) that iterative adversarial training needs models with large capacity for it to be effective. This highlights a limitation of this defense, since it is not clear what model capacity is needed and the models we use already have a large number of parameters.

| MNIST | Baseline | | Gradient Estimation using Finite Differences | | | | Transfer from Model B | | | |
|---|---|---|---|---|---|---|---|---|---|---|
| | | | Single-step | | Iterative | | Single-step | | Iterative | |
| Model | D. of M. | Rand. | FD-xent | FD-logit | IFD-xent | IFD-logit | FGS-xent | FGS-logit | IFGS-xent | IFGS-logit |
| Model A$_{adv\text{-}0.3}$ | 6.5 (5.6) | 1.3 (6.1) | 10.3 (2.6) | 2.8 (5.9) | 36.4 (3.1) | **76.5** (3.1) | 14.6 (6.2) | 14.63 (6.3) | 16.5 (4.7) | 15.9 (4.7) |
| Model A$_{adv\text{-}ens\text{-}0.3}$ | 2.0 (5.6) | 1.2 (6.1) | 6.1 (3.5) | 6.2 (6.3) | 24.2 (4.1) | **96.4** (2.7) | 3.1 (6.2) | 3.1 (6.3) | 4.8 (4.7) | 4.9 (4.7) |
| Model A$_{adv\text{-}iter\text{-}0.3}$ | 3.0 (5.6) | 1.0 (6.1) | 9.2 (7.4) | 7.5 (7.2) | **14.5** (0.96) | 11.6 (3.5) | 11.5 (6.2) | 11.0 (6.3) | 8.7 (4.7) | 8.2 (4.7) |
| **CIFAR-10** | **Baseline** | | **Gradient Estimation using Finite Differences** | | | | **Transfer from Resnet-28-10** | | | |
| | | | Single-step | | Iterative | | Single-step | | Iterative | |
| Model | D. of M. | Rand. | FD-xent | FD-logit | IFD-xent | IFD-logit | FGS-xent | FGS-logit | IFGS-xent | IFGS-logit |
| Resnet-32 $_{adv\text{-}8}$ | 9.6 (440.5) | 10.9 (439.4) | 2.4 (232.9) | 8.5 (401.9) | 69.0 (136.0) | **100.0** (73.8) | 13.1 (439.4) | 13.2 (439.4) | 30.2 (275.4) | 30.2 (275.6) |
| Resnet-32 $_{adv\text{-}ens\text{-}8}$ | 10.1 (440.5) | 10.4 (439.4) | 7.7 (360.2) | 12.2 (399.8) | 95.0 (190.4) | **100.0** (85.2) | 9.7 (439.4) | 9.6 (439.4) | 15.9 (275.4) | 15.5 (275.6) |
| Resnet-32 $_{adv\text{-}iter\text{-}8}$ | 22.86 (440.5) | 21.41 (439.4) | 45.5 (365.5) | 47.5 (331.1) | **55.0** (397.6) | 54.6 (196.3) | 23.2 (439.4) | 23.1 (439.4) | 22.3 (275.4) | 22.3 (275.6) |

Table 3: **Untargeted black-box attacks** for models with **adversarial training**: adversarial success rates and average distortion $\Delta(\mathbf{X}, \mathbf{X}_{adv})$ over the samples. Above: MNIST, $\epsilon = 0.3$. Below: CIFAR-10, $\epsilon = 8$.

**Summary.** Both single-step and iterative variants of the Gradient Estimation attacks outperform other black-box attacks on both the MNIST and CIFAR-10 datasets, achieving attack success rates close to those of white-box attacks even on adversarially trained models, as can be seen in Table 3 and Figure 3.

## 5 ATTACKS ON CLARIFAI: A REAL-WORLD SYSTEM

Since the only requirement for carrying out the Gradient Estimation based attacks is query-based access to the target model, a number of deployed public systems that provide classification as a service can be used to evaluate our methods. We choose Clarifai, as it has a number of models trained to classify image datasets for a variety of practical applications, and it provides black-box access to its models and returns confidence scores upon querying. In particular, Clarifai has models used for the detection of Not Safe For Work (NSFW) content, as well as for Content Moderation. These are important applications where the presence of adversarial samples presents a real danger: an attacker, using query access to the model, could generate an adversarial sample which will no longer be classified as inappropriate. For example, an adversary could upload violent images, adversarially modified, such that they are marked incorrectly as 'safe' by the Content Moderation model.

We evaluate our attack using the Gradient Estimation method on the Clarifai NSFW and Content Moderation models. When we query the API with an image, it returns the confidence scores associated with each category, with the confidence scores summing to 1. We use the *random grouping* technique in order to reduce the number of queries and take the logarithm of the confidence scores in order to use the *logit loss*. A *large number of successful attack images* can be found at `https://www.dropbox.com/s/xsu3l1tjr0yq7rj7/clarifai-examples.zip?dl=0`. Due to their possibly offensive nature, they are not included in the paper.

An example of an attack on the Content Moderation API is given in Figure 1, where the original image on the left is clearly of some kind of drug on a table, with a spoon and a syringe. It is classified as a drug by the Content Moderation model with a confidence score of 0.99. The image on the right is an adversarial image generated with 192 queries to the Content Moderation API, with an $L_\infty$ constraint on the perturbation of $\epsilon = 32$. While the image can still clearly be classified by a human as being of drugs on a table, the Content Moderation model now classifies it as 'safe' with a confidence score of 0.96.

**Remarks.** The proposed Gradient Estimation attacks can successfully generate adversarial examples that are misclassified by a real-world system hosted by Clarifai without prior knowledge of the training set or model.

## 6 CONCLUSION

Overall, in this paper, we conduct a systematic analysis of new and existing black-box attacks on state-of-the-art classifiers and defenses. We propose Gradient Estimation attacks which achieve high attack success rates comparable with even white-box attacks and outperform other state-of-the-art black-box attacks. We apply random grouping and PCA based methods to reduce the number of queries required to a small constant and demonstrate the effectiveness of the Gradient Estimation attack even in this setting. We also apply our black-box attack against a real-world classifier and

state-of-the-art defenses. All of our results show that Gradient Estimation attacks are extremely effective in a variety of settings, making the development of better defenses against black-box attacks an urgent task.

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

## A   EXISTING ATTACKS

In this section, we describe existing methods for generating adversarial examples.

An adversary can generate adversarial example $\mathbf{x}_{\mathrm{adv}}$ from a benign sample $\mathbf{x}$ by adding an appropriate perturbation of small magnitude (Szegedy et al., 2014). Such an adversarial example $\mathbf{x}_{\mathrm{adv}}$ will either cause the classifier to misclassify it into a targeted class (targeted attack), or any class other than the ground truth class (untargeted attack).

### A.1   BLACK-BOX ADVERSARIAL EXAMPLES

Now, we describe two baseline black-box attacks which can be carried out without any knowledge of or query access to the target model.

#### A.1.1   BASELINE ATTACKS

**Random perturbations.** With no knowledge of $f$ or the training set, the simplest manner in which an adversary may seek to carry out an attack is by adding a random perturbation to the input (Szegedy et al., 2014; Goodfellow et al., 2015; Fawzi et al., 2015). These perturbations can be generated by any distribution of the adversary's choice and constrained according to an appropriate norm. If we let $P$ be a distribution over $\mathcal{X}$, and $\mathbf{p}$ is a random variable drawn according to $P$, then a noisy sample is just $\mathbf{x}_{\mathrm{noise}} = \mathbf{x} + \mathbf{p}$. Since random noise is added, it is not possible to generate targeted adversarial samples in a principled manner. This attack is denoted as Rand. throughout.

**Difference of means.** A perturbation aligned with the difference of means of two classes is likely to be effective for an adversary hoping to cause misclassification for a broad range of classifiers (Tramèr et al., 2017b). While these perturbations are far from optimal for DNNs, they provide a useful baseline to compare against. Adversaries with at least partial access to the training or test sets can carry out this attack. An adversarial sample generated using this method, and with $L_\infty$ constraints, is $\mathbf{x}_{\mathrm{adv}} = \mathbf{x} + \epsilon \cdot \mathrm{sign}(\boldsymbol{\mu}_t - \boldsymbol{\mu}_o)$, where $\boldsymbol{\mu}_t$ is the mean of the target class and $\boldsymbol{\mu}_o$ is the mean of the original ground truth class. For an untargeted attack, $t = \mathrm{argmin}_i\, d(\boldsymbol{\mu}_i - \boldsymbol{\mu}_o)$, where $d(\cdot, \cdot)$ is an appropriately chosen distance function. In other words, the class whose mean is closest to the original class in terms of the Euclidean distance is chosen to be the target. This attack is denoted as D. of M. throughout.

#### A.1.2   SINGLE-STEP AND ITERATIVE FAST GRADIENT METHODS

Now, we describe two white-box attack methods, used in transferability-based attacks, for which we constructed approximate, gradient-free versions in Section 3. These attacks are based on either iterative or single-step gradient based minimization of appropriately defined loss functions of neural networks. Since these methods all require the knowledge of the model's gradient, we assume the adversary has access to a local model $f^s$. Adversarial samples generated for $f^s$ can then be transferred to the target model $f^t$ to carry out a *transferability-based attack* (Papernot et al., 2016; Moosavi-Dezfooli et al., 2016). An ensemble of local models (Liu et al., 2017) may also be used. Transferability-based attacks are described in Appendix A.2.

The single-step Fast Gradient method, first introduced by Goodfellow et al. (2015), utilizes a first-order approximation of the loss function in order to construct adversarial samples for the adversary's surrogate local model $f^s$. The samples are constructed by performing a single step of gradient ascent for untargeted attacks. Formally, the adversary generates samples $\mathbf{x}_{\mathrm{adv}}$ with $L_\infty$ constraints (known as the Fast Gradient Sign (FGS) method) in the untargeted attack setting as

$$\mathbf{x}_{\mathrm{adv}} = \mathbf{x} + \epsilon \cdot \mathrm{sign}(\nabla_{\mathbf{x}} \ell_{f^s}(\mathbf{x}, y)), \tag{7}$$

where $\ell_{f^s}(\mathbf{x}, y)$ is the loss function with respect to which the gradient is taken. The loss function typically used is the cross-entropy loss (Goodfellow et al., 2016).

Iterative Fast Gradient methods are simply multi-step variants of the Fast Gradient method described above (Kurakin et al., 2016), where the gradient of the loss is added to the sample for $t + 1$ iterations, starting from the benign sample, and the updated sample is projected to satisfy the constraints $\mathcal{H}$ in every step:

$$\mathbf{x}_{\text{adv}}^{t+1} = \Pi_{\mathcal{H}}(\mathbf{x}_{\text{adv}}^t + \alpha \cdot \text{sign}(\nabla_{\mathbf{x}_{\text{adv}}^t} \ell_{f^s}(\mathbf{x}_{\text{adv}}^t, y))), \tag{8}$$

with $\mathbf{x}_{\text{adv}}^0 = \mathbf{x}$. Iterative fast gradient methods thus essentially carry out projected gradient descent (PGD) with the goal of maximizing the loss, as pointed out by Mądry et al. (2017).

## A.2 TRANSFERABILITY BASED ATTACKS

Here we describe black-box attacks that assume the adversary has access to a representative set of training data in order to train a local model. One of the earliest observations with regards to adversarial samples for neural networks was that they *transfer*; i.e, adversarial attack samples generated for one network are also adversarial for another network. This observation directly led to the proposal of a black-box attack where an adversary would generate samples for a local network and transfer these to the target model, which is referred to as a Transferability based attack.

**Transferability attack (single local model).** These attacks use a *surrogate local model* $f^s$ to craft adversarial samples, which are then submitted to $f$ in order to cause misclassification. Most existing black-box attacks are based on *transferability from a single local model* (Papernot et al., 2016; Moosavi-Dezfooli et al., 2016). The different attack strategies to generate adversarial instances introduced in Section A.1 can be used here to generate adversarial instances against $f^s$, so as to attack $f$.

**Transferability attack (local model ensemble).** Since it is not clear which local model $f^s$ is best suited for generating adversarial samples that transfer well to the target model $f$, Liu et al. (2017) propose the generation of adversarial examples for an ensemble of local models. This method modifies each of the existing transferability attacks by substituting a sum over the loss functions in place of the loss from a single local model.

Concretely, let the ensemble of $m$ local models to be used to generate the local loss be $\{f^{s_1}, \ldots, f^{s_m}\}$. The ensemble loss is then computed as $\ell_{\text{ens}}(\mathbf{x}, y) = \sum_{i=1}^m \alpha_i \ell_{f^{s_i}}(\mathbf{x}, y)$, where $\alpha_i$ is the weight given to each model in the ensemble. The FGS attack in the ensemble setting then becomes $\mathbf{x}_{\text{adv}} = \mathbf{x} + \epsilon \cdot \text{sign}(\nabla_{\mathbf{x}} \ell_{\text{ens}}(\mathbf{x}, y))$. The Iterative FGS attack is modified similarly. Liu et al. (2017) show that the Transferability attack (local model ensemble) performs well even in the targeted attack case, while Transferability attack (single local model) is usually only effective for untargeted attacks. The intuition is that while one model's gradient may not be adversarial for a target model, it is likely that at least one of the gradient directions from the ensemble represents a direction that is somewhat adversarial for the target model.

## B BACKGROUND ON ADVERSARIAL TRAINING

Szegedy et al. (2014) and Goodfellow et al. (2015) introduced the concept of adversarial training, where the standard loss function for a neural network $f$ is modified as follows:

$$\tilde{\ell}(\mathbf{x}, y) = \alpha \ell_f(\mathbf{x}, y) + (1 - \alpha)\ell_f(\mathbf{x}_{\text{adv}}, y), \tag{9}$$

where $y$ is the true label of the sample $\mathbf{x}$. The underlying objective of this modification is to make the neural networks more robust by penalizing it during training to count for adversarial samples. During training, the adversarial samples are computed with respect to the current state of the network using an appropriate method such as FGSM.

**Ensemble adversarial training.** Tramèr et al. (2017a) proposed an extension of the adversarial training paradigm which is called *ensemble adversarial training*. As the name suggests, in ensemble adversarial training, the network is trained with adversarial samples from multiple networks.

**Iterative adversarial training.** A further modification of the adversarial training paradigm proposes training with adversarial samples generated using iterative methods such as the iterative FGSM attack described earlier (Mądry et al., 2017).

# C EVALUATION SETUP DETAILS

## C.1 DATASETS

**MNIST.** This is a dataset of images of handwritten digits (LeCun & Cortes, 1998). There are 60,000 training examples and 10,000 test examples. Each image belongs to a single class from 0 to 9. The images have a dimension $d$ of $28 \times 28$ pixels (total of 784) and are grayscale. Each pixel value lies in $[0, 1]$. The digits are size-normalized and centered. This dataset is used commonly as a 'sanity-check' or first-level benchmark for state-of-the-art classifiers. We use this dataset since it has been extensively studied from the attack perspective by previous work.

**CIFAR-10.** This is a dataset of color images from 10 classes (Krizhevsky & Hinton, 2009). The images belong to 10 mutually exclusive classes (airplane, automobile, bird, cat, deer, dog, frog, horse, ship, and truck). There are 50,000 training examples and 10,000 test examples. There are exactly 6,000 examples in each class. The images have a dimension of $32 \times 32$ pixels (total of 1024) and have 3 channels (Red, Green, and Blue). Each pixel value lies in $[0, 255]$.

## C.2 MODEL TRAINING DETAILS

In this section, we present the architectures and training details for both the normally and adversarially trained variants of the models on both the MNIST and CIFAR-10 datasets. The accuracy of each model on benign data is given in Table 4.

**MNIST.** The model details for the 4 models trained on the MNIST dataset are as follows:

1. Model A (3,382,346 parameters): Conv(64, 5, 5) + Relu, Conv(64, 5, 5) + Relu, Dropout(0.25), FC(128) + Relu, Dropout(0.5), FC + Softmax

2. Model B (710,218 parameters) - Dropout(0.2), Conv(64, 8, 8) + Relu, Conv(128, 6, 6) + Relu, Conv(128, 5, 5) + Relu, Dropout(0.5), FC + Softmax

3. Model C (4,795,082 parameters) - Conv(128, 3, 3) + Relu, Conv(64, 3, 3) + Relu, Dropout(0.25), FC(128) + Relu, Dropout(0.5), FC + Softmax

4. Model D (509,410 parameters) - [FC(300) + Relu, Dropout(0.5)] $\times$ 4, FC + Softmax

Models A and C have both convolutional layers as well as fully connected layers. They also have the same order of magnitude of parameters. Model B, on the other hand, does not have fully connected layers and has an order of magnitude fewer parameters. Similarly, Model D has no convolutional layers and has fewer parameters than all the other models. Models A, B, and C all achieve greater than 99% classification accuracy on the test data. Model D achieves 97.2% classification accuracy, due to the lack of convolutional layers.

For all adversarially trained models, each training batch contains 128 samples of which 64 are benign and 64 are adversarial samples (either FGSM or iterative FGSM). This implies that the loss for each is weighted equally during training; i.e., in Eq. 9, $\alpha$ is set to 0.5. For ensemble adversarial training, the source of the FGSM samples is chosen randomly for each training batch. Networks using standard and ensemble adversarial training are trained for 12 epochs, while those using iterative adversarial training are trained for 64 epochs.

**CIFAR-10.** As their name indicates, Resnet-32 and Resnet-28-10 are ResNet variants (He et al., 2016; Zagoruyko & Komodakis, 2016), while Std.-CNN is a standard CNN (TensorFlow Authors, b). In particular, Resnet-32 is a standard 32 layer ResNet with no width expansion, and Resnet-28-10 is a wide ResNet with 28 layers with the width set to 10, based on the best performing ResNet from Zagoruyko & Komodakis (TensorFlow Authors, a). The width indicates the multiplicative factor by which the number of filters in each residual layer is increased. Std.-CNN is a CNN with two convolutional layers, each followed by a max-pooling and normalization layer and two fully connected layers, each of which has weight decay.

For each model architecture, we train 3 models, one on only the CIFAR-10 training data, one using standard adversarial training and one using ensemble adversarial training. Resnet-32 is trained for 125,000 steps, Resnet-28-10 is trained for 167,000 steps and Std.-CNN is trained for 100,000 steps on the benign training data. Models Resnet-32 and Resnet-28-10 are much more accurate

than Std.-CNN. The adversarial variants of Resnet-32 is trained for 80,000 steps. All models were trained with a batch size of 128.

The two ResNets achieve close to state-of-the-art accuracy ima on the CIFAR-10 test set, with Resnet-32 at 92.4% and Resnet-28-10 at 94.4%. Std.-CNN, on the other hand, only achieves an accuracy of 81.4%, reflecting its simple architecture and the complexity of the task.

Table 4 shows the accuracy of these models with various defenses on benign test data.

| Dataset (Model) | Benign | Adv | Adv-Ens | Adv-Iter |
|---|---|---|---|---|
| MNIST (A) | 99.2 | 99.4 | 99.2 | 99.3 |
| CIFAR-10 (Resnet-32) | 92.4 | 92.1 | 91.7 | 79.1 |

Table 4: Accuracy of models on the benign test data

### C.3 Alternative adversarial success metric

Note that the adversarial success rate can also be computed by considering only the fraction of inputs that meet the adversary's objective *given that the original sample was correctly classified*. That is, one would count the fraction of correctly classified inputs (i.e. $f(\mathbf{x}) = y$) for which $f(\mathbf{x}_{adv}) \neq y$ in the untargeted case, and $f^t(\mathbf{x}_{adv}) = T$ in the targeted case. In a sense, this fraction represents those samples which are truly adversarial, since they are misclassified solely due to the adversarial perturbation added and not due to the classifier's failure to generalize well. In practice, both these methods of measuring the adversarial success rate lead to similar results for classifiers with high accuracy on the test data.

## D  Formal definitions for query-based attacks

Here, we provide a unified framework assuming an adversary can make active queries to the model. Existing attacks making zero queries are a special case in this framework. Given an input instance $\mathbf{x}$, the adversary makes a sequence of queries based on the adversarial constraint set $\mathcal{H}$, and iteratively adds perturbations until the desired query results are obtained, using which the corresponding adversarial example $\mathbf{x}_{adv}$ is generated.

We formally define the targeted and untargeted black-box attacks based on the framework as below.

**Definition 1** (Untargeted black-box attack). *Given an input instance $\mathbf{x}$ and an iterative active query attack strategy $\mathcal{A}$, a query sequence can be generated as $\mathbf{x}^2 = \mathcal{A}(\{(\mathbf{x}^1, q_f^1)\}, \mathcal{H})$, ..., $\mathbf{x}^i = \mathcal{A}(\{(\mathbf{x}^1, q_f^1), \ldots, (\mathbf{x}^{i-1}, q_f^{i-1})\}, \mathcal{H})$, where $q_f^i$ denotes the ith corresponding query result on $\mathbf{x}^i$, and we set $\mathbf{x}^1 = \mathbf{x}$. A black-box attack on $f(\cdot; \theta)$ is untargeted if the adversarial example $\mathbf{x}_{adv} = \mathbf{x}^k$ satisfies $f(\mathbf{x}_{adv}; \theta) \neq f(\mathbf{x}; \theta)$, where $k$ is the number of queries made.*

**Definition 2** (Targeted black-box attack). *Given an input instance $\mathbf{x}$ and an iterative active query attack strategy $\mathcal{A}$, a query sequence can be generated as $\mathbf{x}^2 = \mathcal{A}(\{(\mathbf{x}^1, q_f^1)\}, \mathcal{H})$, ..., $\mathbf{x}^i = \mathcal{A}(\{(\mathbf{x}^1, q_f^1), \ldots, (\mathbf{x}^{i-1}, q_f^{i-1})\}, \mathcal{H})$, where $q_f^i$ denotes the ith corresponding query result on $\mathbf{x}^i$, and we set $\mathbf{x}^1 = \mathbf{x}$. A black-box attack on $f(\cdot; \theta)$ is targeted if the adversarial example $\mathbf{x}_{adv} = \mathbf{x}^k$ satisfies $f(\mathbf{x}_{adv}; \theta) = T$, where $T$ and $k$ are the target class and the number of queries made, respectively.*

The case where the adversary makes no queries to the target classifier is a special case we refer to as a zero-query attack. In the literature, a number of these zero-query attacks have been carried out with varying degrees of success (Papernot et al., 2016; Liu et al., 2017; Moosavi-Dezfooli et al., 2016; Mopuri et al., 2017).

## E  Targeted attacks based on finite differences

The expressions for targeted white-box and Gradient Estimation attacks are given in this section. Targeted transferability attacks are carried out using locally generated targeted white-box adversarial

| | Untageted Transferability to **Model A** | | | |
|---|---|---|---|---|
| | Single-step | | Iterative | |
| Source | FGS-xent | FGS-logit | IFGS-xent | IFGS-logit |
| B | 66.3 (6.2) | 80.8 (6.3) | 89.8 (4.75) | 88.5 (4.75) |
| B,C | 68.1 (6.2) | 89.8 (6.3) | 95.0 (4.8) | 97.1 (4.9) |
| B,C,D | 56.0 (6.3) | 88.7 (6.4) | 73.5 (5.3) | 94.4 (5.3) |
| | Targeted Transferability to **Model A** | | | |
| | Single-step | | Iterative | |
| Source | FGS-T (xent) | FGS-T (logit) | IFGS-T (xent) | IFGS-T (logit) |
| B | 18.3 (6.3) | 18.1 (6.3) | 54.5 (4.6) | 46.5 (4.2) |
| B,C | 23.0 (6.3) | 23.0 (6.3) | 76.7 (4.8) | 72.3 (4.5) |
| B,C,D | 25.2 (6.4) | 25.1 (6.4) | 74.6 (4.9) | 66.1 (4.7) |

Table 5: Adversarial success rates for **transferability-based attacks** on Model A (MNIST) at $\epsilon = 0.3$. Numbers in parentheses beside each entry give the average distortion $\Delta(\mathbf{X}, \mathbf{X}_{\mathrm{adv}})$ over the test set. This table compares the effectiveness of using a single local model to generate adversarial examples versus the use of a local ensemble.

| **MNIST** | **Baseline** | **Gradient Estimation using Finite Differences** | | | | **Transfer from Model B** | | | |
|---|---|---|---|---|---|---|---|---|---|
| | | Single-step | | Iterative | | Single-step | | Iterative | |
| Model | D. of M. | FD-xent | FD-logit | IFD-xent | IFD-logit | FGS-xent | FGS-logit | IFGS-xent | IFGS-logit |
| A | 15.0 (5.6) | 30.0 (6.0) | 29.9 (6.1) | **100.0** (4.2) | 99.7 (2.7) | 18.3 (6.3) | 18.1 (6.3) | 54.5 (4.6) | 46.5 (4.2) |
| B | 35.5 (5.6) | 29.5 (6.3) | 29.3 (6.3) | **99.9** (4.1) | 98.7 (2.4) | - | - | - | - |
| C | 5.84 (5.6) | 34.1 (6.1) | 33.8 (6.4) | **100.0** (4.3) | 99.8 (3.0) | 14.0 (6.3) | 13.8 (6.3) | 34.0 (4.6) | 26.1 (4.2) |
| D | 59.8 (5.6) | 61.4 (6.3) | 60.8 (6.3) | **100.0** (3.7) | 99.9 (1.9) | 16.8 (6.3) | 16.7 (6.3) | 36.4 (4.6) | 32.8 (4.1) |
| **CIFAR-10** | **Baseline** | **Gradient Estimation using Finite Differences** | | | | **Transfer from Resnet-28-10** | | | |
| | | Single-step | | Iterative | | Single-step | | Iterative | |
| Model | D. of M. | FD-xent | FD-logit | IFD-xent | IFD-logit | FGS-xent | FGS-logit | IFGS-xent | IFGS-logit |
| Resnet-32 | 1.2 (440.3) | 23.8 (439.5) | 23.0 (437.0) | **100.0** (110.9) | **100.0** (89.5) | 15.8 (439.4) | 15.5 (439.4) | 71.8 (222.5) | 80.3 (242.6) |
| Resnet-28-10 | 0.9 (440.3) | 29.2 (439.4) | 28.0 (436.1) | 100.0 (123.2) | **100.0** (98.3) | - | - | - | - |
| Std.-CNN | 2.6 (440.3) | 44.5 (439.5) | 40.3 (434.9) | **99.0** (178.8) | 95.0 (126.8) | 5.6 (439.4) | 5.6 (439.4) | 5.1 (222.5) | 5.9 (242.6) |

Table 6: **Targeted black-box attacks**: adversarial success rates. The number in parentheses () for each entry is $\Delta(\mathbf{X}, \mathbf{X}_{\mathrm{adv}})$, the average distortion over all samples used in the attack. Above: MNIST, $\epsilon = 0.3$. Below: CIFAR-10, $\epsilon = 8$.

samples. Adversarial samples generated using the targeted FGS attack are

$$\mathbf{x}_{\mathrm{adv}} = \mathbf{x} - \epsilon \cdot \mathrm{sign}(\nabla_{\mathbf{x}} \ell_{f^s}(\mathbf{x}, T)), \tag{10}$$

where T is the target class. Similarly, the adversarial samples generated using iterative FGS are

$$\mathbf{x}_{\mathrm{adv}}^{t+1} = \Pi_{\mathcal{H}}(\mathbf{x}_{\mathrm{adv}}^t - \alpha \cdot \mathrm{sign}(\nabla_{\mathbf{x}_{\mathrm{adv}}^t} \ell_{f^s}(\mathbf{x}_{\mathrm{adv}}^t, T))). \tag{11}$$

For the logit based loss, targeted adversarial samples are generated using the following loss term:

$$\mathbf{x}_{\mathrm{adv}} = \mathbf{x} - \epsilon \cdot \mathrm{sign}(\nabla_{\mathbf{x}}(\max(\boldsymbol{\phi}(\mathbf{x})_i : i \neq T) - \boldsymbol{\phi}(\mathbf{x})_T)). \tag{12}$$

Targeted black-box adversarial samples generated using the Gradient Estimation method are then

$$\mathbf{x}_{\mathrm{adv}} = \mathbf{x} - \epsilon \cdot \mathrm{sign}\left( \frac{\mathrm{FD}_{\mathbf{x}}(p_T^f(\mathbf{x}), \delta)}{p_T^f(\mathbf{x})} \right). \tag{13}$$

Similarly, in the case of a black-box adversary with query-access to the logits, the adversarial sample is

$$\mathbf{x}_{\mathrm{adv}} = \mathbf{x} - \epsilon \cdot \mathrm{sign}(\mathrm{FD}_{\mathbf{x}}(\max(\boldsymbol{\phi}(\mathbf{x})_i : i \neq T) - \boldsymbol{\phi}(\mathbf{x})_T, \delta)). \tag{14}$$

# F   GRADIENT ESTIMATION WITH QUERY REDUCTION

## F.1   RANDOM GROUPING

This section contains the detailed algorithm for query reduction using random grouping.

---

**Algorithm 1** Gradient estimation with query reduction using random features

---

**Input:** $\mathbf{x}, k, \delta, g(\cdot)$

**Output:** Estimated gradient $\hat{\nabla}_{\mathbf{x}} g(\mathbf{x})$ of $g(\cdot)$ at $\mathbf{x}$

  1: Initialize empty vector $\hat{\nabla}_{\mathbf{x}} g(\mathbf{x})$ of dimension $d$

  2: **for** $i \leftarrow 1$ to $\lceil \frac{d}{k} \rceil - 1$ **do**

  3:      Choose a set of random $k$ indices $S_i$ out of $[1, \ldots, d]/\{\cup_{j=1}^{i-1} S_j\}$

  4:      Initialize $\mathbf{v}$ such that $\mathbf{v}_j = 1$ iff $j \in S_i$

  5:      For all $j \in S_i$, set $\hat{\nabla}_{\mathbf{x}} g(\mathbf{x})_j = \frac{g(\mathbf{x}+\delta\mathbf{v})-g(\mathbf{x}-\delta\mathbf{v})}{2\delta k}$, which is the two-sided approximation of the directional derivative along $\mathbf{v}$

  6: **end for**

  7: Initialize $\mathbf{v}$ such that $\mathbf{v}_j = 1$ iff $j \in [1, \ldots, d]/\{\cup_{j=1}^{\lceil \frac{d}{k} \rceil - 1} S_j\}$

  8: For all $j \in [1, \ldots, d]/\{\cup_{j=1}^{\lceil \frac{d}{k} \rceil - 1} S_j\}$, set $\hat{\nabla}_{\mathbf{x}} g(\mathbf{x})_j = \frac{g(\mathbf{x}+\delta\mathbf{v})-g(\mathbf{x}-\delta\mathbf{v})}{2\delta k}$

---

## F.2 PCA

Concretely, let the samples the adversary wants to misclassify be column vectors $\mathbf{x}^i \in \mathbb{R}^d$ for $i \in \{1, \ldots, n\}$ and let $\mathbf{X}$ be the $d \times n$ matrix of centered data samples (i.e. $\mathbf{X} = [\tilde{\mathbf{x}}^1 \tilde{\mathbf{x}}^2 \ldots \tilde{\mathbf{x}}^n]$, where $\tilde{\mathbf{x}}^i = \mathbf{x} - \frac{1}{n} \sum_{j=1}^n \mathbf{x}^j$). The principal components of $\mathbf{X}$ are the normalized eigenvectors of its sample covariance matrix $\mathbf{C} = \mathbf{X}\mathbf{X}^{\mathsf{T}}$. Since $\mathbf{C}$ is a positive semidefinite matrix, there is a decomposition $\mathbf{C} = \mathbf{U}\Lambda\mathbf{U}^{\mathsf{T}}$ where $\mathbf{U}$ is an orthogonal matrix, $\Lambda = \mathrm{diag}(\lambda_1, \ldots, \lambda_d)$, and $\lambda_1 \geq \ldots \geq \lambda_d \geq 0$. Thus, $\mathbf{U}$ in Algorithm 2 is the $d \times d$ matrix whose columns are unit eigenvectors of $\mathbf{C}$. The eigenvalue $\lambda_i$ is the variance of $\mathbf{X}$ along the $i^{\text{th}}$ component. Further, PCA minimizes reconstruction error in terms of the $L_2$ norm; i.e., it provides a basis in which the Euclidean distance to the original sample from a sample reconstructed using a subset of the basis vectors is the smallest.

---

**Algorithm 2** Gradient estimation with query reduction using PCA components

---

**Input:** $\mathbf{x}, k, \mathbf{U}, \delta, g(\cdot)$

**Output:** Estimated gradient $\hat{\nabla}_{\mathbf{x}} g(\mathbf{x})$ of $g(\cdot)$ at $\mathbf{x}$

  1: **for** $i \leftarrow 1$ to $k$ **do**

  2:      Initialize $\mathbf{v}$ such that $\mathbf{v} = \frac{\mathbf{u}_i}{\|\mathbf{u}_i\|}$, where $\mathbf{u}_i$ is the $i^{\text{th}}$ column of $\mathbf{U}$

  3:      Compute

$$\alpha^i(\mathbf{v}) = \frac{g(\mathbf{x} + \delta\mathbf{v}) - g(\mathbf{x} - \delta\mathbf{v})}{2\delta},$$

        which is the two-sided approximation of the directional derivative along $\mathbf{v}$

  4:      Update $\hat{\nabla}_{\mathbf{x}} g(\mathbf{x})^i = \hat{\nabla}_{\mathbf{x}} g(\mathbf{x})^{i-1} + \alpha^i(\mathbf{v})\mathbf{v}$

  5: **end for**

  6: Set $\hat{\nabla}_{\mathbf{x}} g(\mathbf{x}) = \hat{\nabla}_{\mathbf{x}} g(\mathbf{x})^k$

---

## G  SUMMARY OF ATTACKS EVALUATED

**Taxonomy of black-box attacks**: To deepen our understanding of the effectiveness of black-box attacks, in this work, we propose a taxonomy of black-box attacks, intuitively based on the number of queries on the target model used in the attack. The details are provided in Table 7.

We evaluate the following attacks summarized in Table 7:

    1. Zero-query attacks

        (a) Baseline attacks: Random-Gaussian perturbations (Rand.) and Difference-of-Means aligned perturbations (D. of M.)

        (b) Transferability attack (single local model) using Fast Gradient Sign (FGS) and Iterative FGS (IFGS) samples generated on a single source model for both loss functions (Transfer *model* FGS/IFGS-*loss*); e.g., Transfer Model A FGS-logit

| | | Attack | | | | Abbreviation | Untargeted | Targeted |
|---|---|---|---|---|---|---|---|---|
| Black-box | Zero-query | Random-Gaussian-perturbation | | | | Rand. | + | |
| | | Difference-of-means | | | | D. of M. | + | + |
| | | Transfer from *model* | Surrogate attack | Steps | Loss function | | | |
| | | | FGS | Single-step | Cross-entropy | Transfer *model* FGS-xent | + | + |
| | | | | | Logit-based | Transfer *model* FGS-logit | + | + |
| | | | IFGS | Iterative | Cross-entropy | Transfer *model* IFGS-xent | + | + |
| | | | | | Logit-based | Transfer *model* IFGS-logit | + | + |
| | Query based | Finite-difference gradient estimation | | Steps | Loss function | | | |
| | | | | Single-step | Cross-entropy | FD-xent | + | + |
| | | | | | Logit-based | FD-logit | + | + |
| | | | | Iterative | Cross-entropy | IFD-xent | + | + |
| | | | | | Logit-based | IFD-logit | + | + |
| | | Query-reduced gradient estimation | Technique | Steps | Loss function | | | |
| | | | Random grouping | Single-step | Logit-based | GE-QR (RG-$k$, logit) | + | + |
| | | | | Iterative | Logit-based | IGE-QR (RG-$k$, logit) | + | + |
| | | | PCA | Single-step | Logit-based | GE-QR (PCA-$k$, logit) | + | + |
| | | | | Iterative | Logit-based | IGE-QR (PCA-$k$, logit) | + | + |
| White-box | | Fast gradient sign (FGS) | | Steps | Loss function | | | |
| | | | | Single-step | Cross-entropy | WB FGS-xent | + | + |
| | | | | | Logit-based | WB FGS-logit | + | + |
| | | | | Iterative | Cross-entropy | WB IFGS-xent | + | + |
| | | | | | Logit-based | WB IFGS-logit | + | + |

Table 7: Attacks evaluated in this paper

    (c) Transferability attack (local model ensemble) using FGS and IFGS samples generated on a source model for both loss functions (Transfer *models* FGS/IFGS-*loss*); e.g., Transfer Model B, Model C IFGS-logit

2. Query based attacks

    (a) Finite-difference and Iterative Finite-difference attacks for the gradient estimation attack for both loss functions (FD/IFD-*loss*); e.g., FD-logit

    (b) Gradient Estimation and Iterative Gradient Estimation with Query reduction attacks (IGE/GE-QR (*Technique*-$k$, *loss*)) using two query reduction techniques, random grouping (RG) and principal component analysis components (PCA); e.g., GE-QR (PCA-$k$, logit)

3. White-box FGS and IFGS attacks for both loss functions (WB FGS/IFGS (*loss*))

## H  ADVERSARIAL SAMPLES

In Figure 4, we show some examples of successful untargeted adversarial samples against Model A on MNIST and Resnet-32 on CIFAR-10. These images were generated with an $L_\infty$ constraint of $\epsilon = 0.3$ for MNIST and $\epsilon = 8$ for CIFAR-10. Clearly, the amount of perturbation added by iterative attacks is much smaller, barely being visible in the images.

## I  DETAILED EVALUATION RESULTS

### I.1  WHITE-BOX ATTACK RESULTS

In this section, we present the white-box attack results for various cases in Tables 8–10. Where relevant, our results match previous work (Goodfellow et al., 2015; Kurakin et al., 2016).

### I.2  EFFECTIVENESS OF BASELINE ATTACKS

In the baseline attacks described in Appendix A.1.1, the choice of distribution for the random perturbation attack and the choice of distance function for the difference of means attack are not fixed. Here, we describe the choices we make for both attacks. The random perturbation $\mathbf{p}$ for each sample (for both MNIST and CIFAR-10) is chosen independently according to a multivariate normal distribution with mean $\mathbf{0}$, i.e. $\mathbf{p} \sim \mathcal{N}(\mathbf{0}, \mathbf{I}_d)$. Then, depending on the norm constraint, either a signed and scaled version of the random perturbation ($L_\infty$) or a scaled unit vector in the direction of the perturbation ($L_2$) is added. For an untargeted attack utilizing perturbations aligned with the difference of means, for each sample, the mean of the class closest to the original class in the $L_2$ distance is determined.

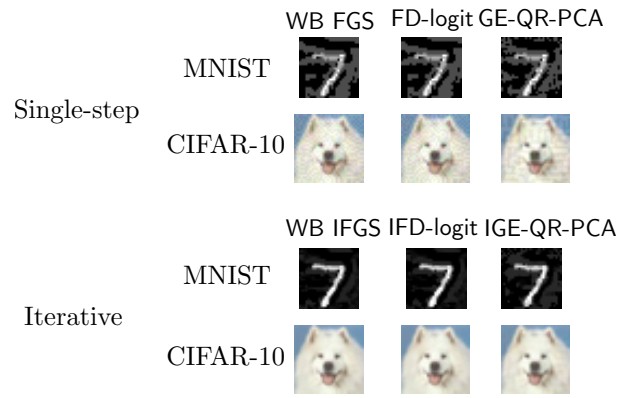

Figure 4: Untargeted **adversarial samples** on Model A on MNIST and Resnet-32 on CIFAR-10. All attacks use the logit loss. Perturbations in the images generated using single-step attacks are far smaller than those for iterative attacks. The '7' from MNIST is classified as a '3' by all single-step attacks and as a '9' by all iterative attacks. The *dog* from CIFAR-10 is classified as a *bird* by the white-box FGS and Finite Difference attack, and as a *frog* by the Gradient Estimation attack with query reduction.

| **MNIST** | **White-box** | | | |
|---|---|---|---|---|
| | Single-step | | Iterative | |
| Model | FGS (xent) | FGS (logit) | IFGS (xent) | IFGS (logit) |
| A | 69.2 (5.9) | 90.1 (5.9) | 99.5 (4.4) | **100.0** (2.1) |
| B | 84.7 (6.2) | 98.8 (6.3) | **100.0** (4.75) | **100.0** (1.6) |
| C | 67.9 (6.1) | 76.5 (6.6) | **100.0** (4.7) | **100.0** (2.2) |
| D | 98.3 (6.3) | **100.0** (6.5) | **100.0** (5.6) | **100.0** (1.2) |

| **CIFAR-10** | **White-box** | | | |
|---|---|---|---|---|
| | Single-step | | Iterative | |
| Model | FGS (xent) | FGS (logit) | IFGS (xent) | IFGS (logit) |
| Resnet-32 | 82.6 (439.7) | 86.8 (438.3) | **100.0** (247.2) | **100.0** (66.1) |
| Resnet-28-10 | 86.4 (439.6) | 87.4 (439.2) | **100.0** (278.7) | **100.0** (75.4) |
| Std.-CNN | 93.9 (439.6) | 98.5 (429.8) | 98.0 (314.3) | **100.0** (80.3) |

Table 8: **Untargeted white-box attacks**: adversarial success rates and average distortion $\Delta(\mathbf{X}, \mathbf{X}_{\mathrm{adv}})$ over the test set. Above: MNIST, $\epsilon = 0.3$. Below: CIFAR-10, $\epsilon = 8$.

As expected, adversarial samples generated using Rand. do not achieve high adversarial success rates in spite of having similar or larger average distortion than the other black-box attacks for both the MNIST and CIFAR-10 models. However, the D. of M. method is quite effective at higher perturbation values for the MNIST dataset as can be seen in Figure 2a. Also, for Models B and D, the D. of M. attack is more effective than FD-xent. The D. of M. method is less effective in the targeted attack case, but for Model D, it outperforms the transferability based attack considerably. Its success rate is comparable to the targeted transferability based attack for Model A as well.

The relative effectiveness of the two baseline methods is reversed for the CIFAR-10 dataset, however, where Rand. outperforms D. of M. considerably as $\epsilon$ is increased. This indicates that the models trained on MNIST have normal vectors to decision boundaries which are more aligned with the vectors along the difference of means as compared to the models on CIFAR-10.

### I.3 TRANSFERABILITY ATTACK RESULTS

For the transferability experiments, we choose to transfer from Model B for MNIST dataset and from Resnet-28-10 for CIFAR-10 dataset, as these models are each similar to at least one of the

| MNIST | White-box | | | |
|---|---|---|---|---|
| | Single-step | | Iterative | |
| Model | FGS (xent) | FGS (logit) | IFGS (xent) | IFGS (logit) |
| A | 30.1 (6.1) | 30.1 (6.1) | **100.0** (4.7) | 99.6 (2.7) |
| B | 29.6 (6.2) | 29.4 (6.3) | **100.0** (4.6) | 98.7 (2.4) |
| C | 33.2 (6.4) | 33.0 (6.4) | **100.0** (4.6) | 99.8 (3.0) |
| D | 61.5 (6.4) | 60.9 (6.3) | **100.0** (4.7) | 99.9 (2.0) |
| **CIFAR-10** | **White-box** | | | |
| | Single-step | | Iterative | |
| Model | FGS (xent) | FGS (logit) | IFGS (xent) | IFGS (logit) |
| Resnet-32 | 23.7 (439.6) | 23.5 (436.0) | **100.0** (200.4) | **100.0** (89.5) |
| Resnet-28-10 | 28.0 (439.6) | 27.6 (436.5) | **100.0** (215.7) | **100.0** (99.0) |
| Std.-CNN | 43.8 (439.5) | 40.2 (435.6) | **99.0** (262.4) | 95.0 (127.8) |

Table 9: **Targeted white-box attacks**: adversarial success rates and average distortion $\Delta(\mathbf{X}, \mathbf{X}_{\text{adv}})$ over the test set. Above: MNIST, $\epsilon = 0.3$. Below: CIFAR-10, $\epsilon = 8$.

| MNIST | White-box | | | |
|---|---|---|---|---|
| | Single-step | | Iterative | |
| Model | FGS (xent) | FGS (logit) | IFGS (xent) | IFGS (logit) |
| Model A$_{\text{adv-0.3}}$ | 2.8 (6.1) | 2.9 (6.0) | **79.1** (4.2) | 78.5 (3.1) |
| Model A$_{\text{adv-ens-0.3}}$ | 6.2 (6.2) | 4.6 (6.3) | 93.0 (4.1) | **96.2** (2.7) |
| Model A$_{\text{adv-iter-0.3}}$ | 7.0 (6.4) | 5.9 (7.5) | 10.8 (3.6) | **11.0** (3.6) |
| **CIFAR-10** | **White-box** | | | |
| | Single-step | | Iterative | |
| Model | FGS (xent) | FGS (logit) | IFGS (xent) | IFGS (logit) |
| Resnet-32 $_{\text{adv-8}}$ | 1.1 (439.7) | 1.5 (438.8) | **100.0** (200.6) | **100.0** (73.7) |
| Resnet-32 $_{\text{adv-ens-8}}$ | 7.2 (439.7) | 6.6 (437.9) | **100.0** (201.3) | **100.0** (85.3) |
| Resnet-32 $_{\text{adv-iter-8}}$ | 48.3 (438.2) | 50.4 (346.6) | 54.9 (398.7) | **57.3** (252.4) |

Table 10: **Untargeted white-box attacks for models with adversarial training**: adversarial success rates and average distortion $\Delta(\mathbf{X}, \mathbf{X}_{\text{adv}})$ over the test set. Above: MNIST, $\epsilon = 0.3$. Below: CIFAR-10, $\epsilon = 8$.

other models for their respective dataset and different from one of the others. They are also fairly representative instances of DNNs used in practice.

Adversarial samples generated using single-step methods and transferred from Model B to the other models have higher success rates for untargeted attacks when they are generated using the logit loss as compared to the cross entropy loss as can be seen in Table 1. For iterative adversarial samples, however, the untargeted attack success rates are roughly the same for both loss functions. As has been observed before, the adversarial success rate for targeted attacks with transferability is much lower than the untargeted case, even when iteratively generated samples are used. For example, the highest targeted transferability rate in Table 6 is 54.5%, compared to 100.0% achieved by IFD-xent-T across models. One attempt to improve the transferability rate is to use an ensemble of local models, instead of a single one. The results for this on the MNIST data are presented in Table 5. In general, both untargeted and targeted transferability increase when an ensemble is used. However, the increase is not monotonic in the number of models used in the ensemble, and we can see that the transferability rate for IFGS-xent samples falls sharply when Model D is added to the ensemble. This may be due to it having a very different architecture as compared to the models, and thus also having very different gradient directions. This highlights one of the pitfalls of transferability, where it is important to use a local surrogate model similar to the target model for achieving high attack success rates.

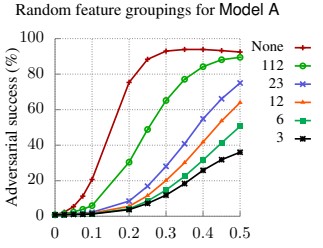 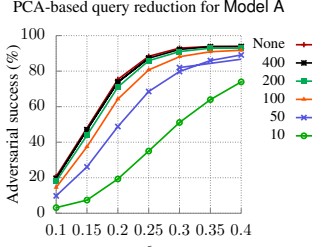 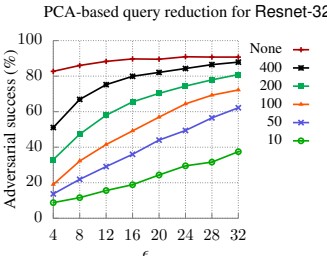

(a) Gradient Estimation attack with query reduction using random grouping and the logit loss (GE-QR (RG-$k$, logit)) on Model A (MNIST, $d = 784$). The adversarial success rate decreases as the number of groups $\lceil \frac{d}{k} \rceil$ is decreased, where $k$ is the size of the group and $d$ is the dimension of the input.

(b) Gradient Estimation attack with query reduction using PCA components and the logit loss (GE-QR (PCA-$k$, logit)) on Model A (MNIST, $d = 784$). The adversarial success rates decrease as the number of principal components $k$ used for estimation is decreased. Relatively high success rates are maintained even for $k = 50$.

(c) Gradient Estimation attack with query reduction using PCA components and the logit loss (GE-QR (PCA-$k$, logit)) on Resnet-32 (CIFAR-10). Relatively high success rates are maintained even for $k = 400$.

Figure 5: **Adversarial success rates for Gradient Estimation attacks with query reduction (FD-QR (*Technique*, logit)) on Model A (MNIST) and Resnet-32 (CIFAR-10)**, where *Technique* is either PCA or *RG*. 'None' refers to FD-logit, the case where the number of queries is $2d$, where $d$ is the dimension of the input.

## I.4    EFFECT OF DIMENSION ON GRADIENT ESTIMATION ATTACKS WITH QUERY REDUCTION

We consider the effectiveness of Gradient Estimation with random grouping based query reduction and the logit loss (GE-QR (RG-$k$, logit)) on Model A on MNIST data in Figure 5a, where $k$ is the number of indices chosen in each iteration of Algorithm 1. Thus, as $k$ increases and the number of groups decreases, we expect adversarial success to decrease as gradients over larger groups of features are averaged. This is the effect we see in Figure 5a, where the adversarial success rate drops from 93% to 63% at $\epsilon = 0.3$ as $k$ increases from 1 to 7. Grouping with $k = 7$ translates to 112 queries per MNIST image, down from 784. Thus, in order to achieve high adversarial success rates with the random grouping method, larger perturbation magnitudes are needed.

On the other hand, the PCA-based approach GE-QR (PCA-$k$, logit) is much more effective, as can be seen in Figure 5b. Using 100 principal components to estimate the gradient for Model A on MNIST as in Algorithm 2, the adversarial success rate at $\epsilon = 0.3$ is 88.09%, as compared to 92.9% without any query reduction. Similarly, using 400 principal components for Resnet-32 on CIFAR-10 (Figure 5c), an adversarial success rate of 66.9% can be achieved at $\epsilon = 8$. At $\epsilon = 16$, the adversarial success rate rises to 80.1%.

## I.5    SINGLE-STEP ATTACKS ON DEFENSES

In this section, we analyse the effectiveness of single-step black-box attacks on adversarially trained models and show that the Gradient Estimation attacks using Finite Differences with the addition of random perturbations outperform other black-box attacks.

**Evaluation of single-step attacks on model with basic adversarial training**: In Figure 6a, we can see that both single-step black-box and white-box attacks have much lower adversarial success rates on Model A$_{adv-0.3}$ as compared to Model A. The success rate of the Gradient Estimation attacks matches that of white-box attacks on these adversarially trained networks as well. To overcome this, we add an initial random perturbation to samples before using the Gradient Estimation attack with Finite Differences and the logit loss (FD-logit). These are then the most effective single step black-box attacks on Model A$_{adv-0.3}$ at $\epsilon = 0.3$ with an adversarial success rate of 32.2%, surpassing the Transferability attack (single local model) from B.

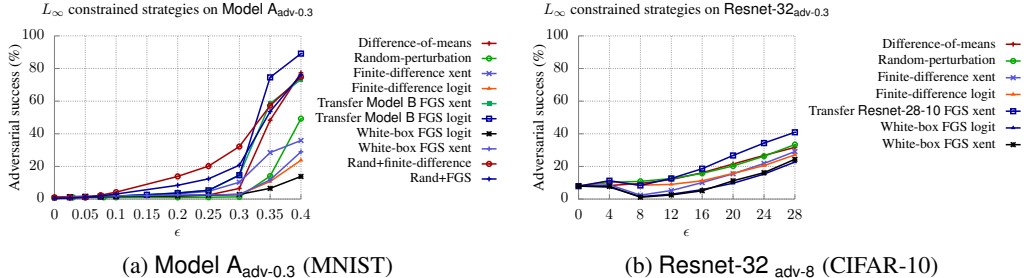

Figure 6: **Effectiveness of various single step black-box attacks against adversarially trained models**. On the MNIST model, Model A$_{adv-0.3}$ the attack with the highest performance up till $\epsilon = 0.3$ is the Gradient Estimation attack using Finite Differences with initially added randomness. Beyond this, the Transferability attack (single local model) using samples from Model B performs better. On the CIFAR-10 model Resnet-32 $_{adv-8}$, the best performing attack is the Transferability attack (single local model) using samples from Resnet-28-10.

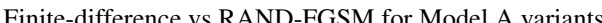

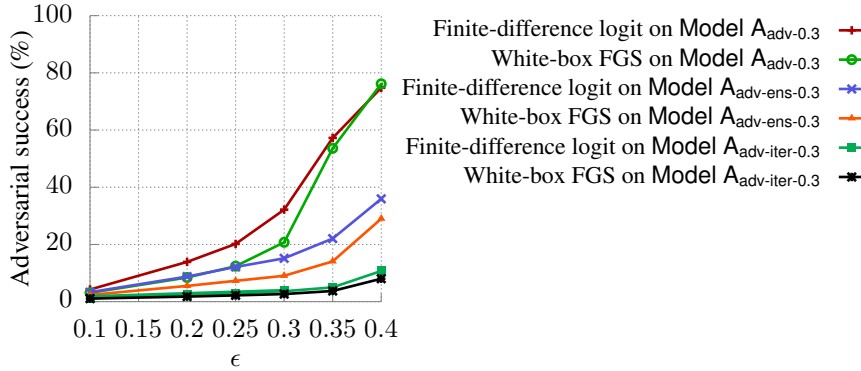

Figure 7: **Increasing the effectiveness of FD-logit attacks on Model A$_{adv-0.3}$, Model A$_{adv-ens-0.3}$ and Model A$_{adv-iter-0.3}$ (MNIST) by adding an initial $L_\infty$ constrained random perturbation of magnitude 0.01.**

In Figure 6b, we again see that the Gradient Estimation attacks using Finite Differences (FD-xent and FD-logit) and white-box FGS attacks (FGS-xent and FGS-logit) against Resnet-32. As $\epsilon$ is increased, the attacks that perform the best are Random Perturbations (Rand.), Difference-of-means (D. of M.), and Transferability attack (single local model) from Resnet-28-10 with the latter performing slightly better than the baseline attacks. This is due to the 'gradient masking' phenomenon and can be overcome by adding random perturbations as for MNIST. An interesting effect is observed at $\epsilon = 4$, where the adversarial success rate is *higher* than at $\epsilon = 8$. The likely explanation for this effect is that the model has overfitted to adversarial samples at $\epsilon = 8$. Our Gradient Estimation attack closely tracks the adversarial success rate of white-box attacks in this setting as well.

**Increasing effectiveness of single-step attacks using initial random perturbation**: Since the Gradient Estimation attack with Finite Differences (FD-xent and FD-logit) were not performing well due the masking of gradients at the benign sample $\mathbf{x}$, we added an initial random perturbation to escape this low-gradient region as in the RAND-FGSM attack (Tramèr et al., 2017a). Figure 7 shows the effect of adding an initial $L_\infty$-constrained perturbation of magnitude 0.05. With the addition of a random perturbation, FD-logit has a much improved adversarial success rate on Model A$_{adv-0.3}$, going up to 32.2% from 2.8% without the perturbation at a total perturbation value of 0.3. It even outperforms the white-box FGS (FGS-logit) with the same random perturbation added. This effect is also observed for Model A$_{adv-ens-0.3}$, but Model A$_{adv-iter-0.3}$ appears to be resistant to single-step

gradient based attacks. Thus, our attacks work well for single-step attacks on DNNs with standard and ensemble adversarial training, and achieve performance levels close to that of white-box attacks.

## I.6 EFFICIENCY OF GRADIENT ESTIMATION ATTACKS

In our evaluations, all models were run on a GPU with a batch size of 100. On Model A on MNIST data, single-step attacks FD-xent and FD-logit take $6.2 \times 10^{-2}$ and $8.8 \times 10^{-2}$ seconds per sample respectively. Thus, these attacks can be carried out on the entire MNIST test set of 10,000 images in about 10 minutes. For iterative attacks with no query reduction, with 40 iterations per sample ($\alpha$ set to 0.01), both IFD-xent and IFD-xent-T taking about 2.4 seconds per sample. Similarly, IFD-logit and IFD-logit-T take about 3.5 seconds per sample. With query reduction, using IGE-QR (PCA-$k$, logit) with $k = 100$ and IGE-QR (RG-$k$, logit) with $k = 8$, the time taken is just 0.5 seconds per sample. In contrast, the fastest attack from Chen et al. (2017), the ZOO-ADAM attack, takes around 80 seconds per sample for MNIST, which is $24\times$ slower than the Iterative Finite Difference attacks and around $160\times$ slower than the Iterative Gradient Estimation attacks with query reduction.

For Resnet-32 on the CIFAR-10 dataset, FD-xent, FD-xent-T, FD-logit and FD-logit-T all take roughly 3s per sample. The iterative variants of these attacks with 10 iterations ($\alpha$ set to 1.0) take roughly 30s per sample. Using query reduction, both IGE-QR (PCA-$k$, logit) with $k = 100$ with 10 iterations takes just 5s per sample. The time required per sample increases with the complexity of the network, which is observed even for white-box attacks. For the CIFAR-10 dataset, the fastest attack from Chen et al. (2017) takes about 206 seconds per sample, which is $7\times$ slower than the Iterative Finite Difference attacks and around $40\times$ slower than the Iterative Gradient Estimation attacks with query reduction.

All the above numbers are for the case when queries are not made in parallel. Our attack algorithm allows for queries to be made in parallel as well. We find that a simple parallelization of the queries gives us a $2 - 4\times$ speedup. The limiting factor is the fact that the model is loaded on a single GPU, which implies that the current setup is not fully optimized to take advantage of the inherently parallel nature of our attack. With further optimization, greater speedups can be achieved.

**Remarks**: Overall, our attacks are very efficient and allow an adversary to generate a large number of adversarial samples in a short period of time.

