# OpenReview forum: "Exploring the Space of Black-box Attacks on Deep Neural Networks"
_ICLR.cc/2018/Conference — Reject_

### Official Review · AnonReviewer3 · 2017-11-27
**Empirical study of standard adversarial attacks + basic gradient approximation methods**

**Rating:** 5
**Confidence:** 4

**Review:**

This paper generates adversarial examples using the fast gradient sign (FGS) and iterated fast gradient sign (IFGS) methods, but replacing the gradient computation with finite differences or another gradient approximation method. Since finite differences is expensive in high dimensions, the authors propose using directional derivatives based on random feature groupings or PCA.

This paper would be much stronger if it surveyed a wider variety of gradient-free optimization methods. Notably, there's two important black-box optimization baselines that were not included: simultaneous perturbation stochastic approximation ( https://en.wikipedia.org/wiki/Simultaneous_perturbation_stochastic_approximation), which avoids computing the gradient explicitly, and evolutionary strategies ( https://blog.openai.com/evolution-strategies/ ), a similar method that uses several random directions to estimate a better descent direction.

The gradient approximation methods proposed in this paper may or may not be better than SPSA or ES. Without a direct comparison, it's hard to know.  Thus, the main contribution of this paper is in demonstrating that gradient approximation methods are sufficient for generating good adversarial attacks and applying those attacks to Clarifai models. That's interesting and useful to know, but is still a relatively small contribution, making this paper borderline. I lean towards rejection, since the paper proposes new methods without comparing to or even mentioning well-known alternatives.

REVISION: Thank you for your response! The additional material does strengthen the paper. There is now some discussion of how Chen et al. differs, and an explicit comparison to SPSA and PSO. I think there are some interesting results here, including attacks on Clarifai. However, the additional evaluations are not thorough. This is understandable (given the limited time frame), but unfortunate. SPSA is only evaluated on MNIST, and while the paper claims its distortion is greater, this is never shown explicitly (or was too difficult for me to find, even when searching through the revised paper). Chen et al. is only compared in terms of time, not on success rate, distortion, or number of queries. These timing results aren't necessarily comparable, since the experiments were done under different conditions. Overall, the new experiments and discussion are a step towards a thorough analysis of zero-order attacks, but they're not there yet. I've increased my rating from 4 to 5, but this is still below the bar for me.

---

> ### Author Response · Authors · 2017-12-20
> **Surveying a broader range of gradient-free optimization methods in updated version**
>
> We thank the reviewer for the insightful suggestions. We did experiment with other gradient-free optimization methods including Particle Swarm Optimization (PSO) and Simultaneous perturbation stochastic approximation (SPSA). We found that the Gradient Estimation method achieves the highest attack success rates and due to the limitation of space, these results were not included in the initial submitted version. In the updated version, we have added Section 3.4 which contains a discussion of these two methods. Table 2 contains a quantitative comparative evaluation of all the types of query-based black-box attacks we tried.
>
> We found Particle Swarm Optimization to perform very poorly as it ran much slower than the other methods without achieving success rates comparable to the other methods. SPSA ran faster and was able to achieve attack success rates matching those of the Gradient Estimation attacks. However, we had to experiment with a large number of parameters in order for SPSA to be effective, unlike Gradient Estimation which required very little parameter tuning. Further, the adversarial samples found using SPSA had almost twice as high distortion as those found using Gradient Estimation based attacks.

---

> ### Author Response · Authors · 2018-01-18
> **Regarding higher distortion for SPSA**
>
> Thank you for your revised review. Regarding the higher value of distortion for SPSA, we would like to refer you to the second column of Table 2 titled 'Attack success'. The numbers in parentheses in this column provide the average distortion value for each type of attack. Since the earlier table (Table 1) of results had mentioned that numbers in parentheses represent average distortion, we omitted that detail here. We apologize for the confusion.

---

### Official Review · AnonReviewer1 · 2017-11-28
**About assumptions and metrics**

**Rating:** 6
**Confidence:** 3

**Review:**


Quality: The paper studies an important problem given that public ML APIs are now becoming available. More specifically, the authors study black-box attacks based on gradient estimation. This means that adversaries have no access to the underlying model.

Clarity: The paper is clear and well-written. Some parts are a bit redundant, so more space of the main body of the paper could be devoted information provided in the appendix and would help with the flow (e.g., description of the models A, B, C; logit-based loss; etc.). This would also provide room for discussing the targeted attacks and the tranferability-based attacks.

Originality: While black-box attacks are of greater interest than withe-box attacks, I found the case considered here of modest interest. The assumption that the loss would be known, but not the gradient is relatively narrow. And why is not possible to compute the gradient exactly in this case? Also, it was not clear what how \delta can be chosen in practice to increase the performance of the attack. Could the authors comment on that?

Significance: The results in the paper are encouraging, but it is not clear whether the setting is realistic. The main weakness of this paper is that it does not state the assumptions made and under which conditions these attacks are valid. Those have to be deduced from the main text and not all are clear and many questions remain, making it difficult to see when such an attack is a risk and what is the actual experimental set-up. For example, what does it mean that attackers have access to the training set and when does that occur? Is it assumed that the API uses the adversarial example for training as well or not? How are the surrogate models trained and what are they trying to optimize and/or what do they match? In which situations do attackers have access to the loss, but not the gradient? How sensitive are the results to a loss mismatch? Finally, I do not understand the performance metric proposed by the authors. It is always possible to get an arbitrarily high success rate unless one fixes the distortion. What would be the success rate if the distortion was equal to the distortion of white-box attacks?  And how sensitive are the results to \epsilon (and how can it be chosen by an attacker in practice)?

---

> ### Author Response · Authors · 2017-12-20
> **Clarifying assumptions, metrics and problem setting**
>
> We thank the reviewer for the nice suggestions. We provide details for aspects of the paper the reviewer found unclear and update the paper for clarity.
>
> In the black-box threat model considered in our paper, the model is not known, and without access to the model, automatic differentiation methods cannot be used to obtain the true gradient by backpropagation. Attackers would have access only to the loss and not the gradient of the model if they were able to query the target model for its classification output, consisting of class probabilities, but did not have access to the model itself. Both the cross-entropy and logit losses we consider can be easily computed from the output probabilities, which are provided for deployed classification systems by a number of MLaaS companies.
> Since an estimate of the true gradient of a function can be calculated with access to just the function values, we compute the loss from the output probabilities and use this to estimate the gradient of the loss. The estimate of the gradient of the loss is then used to compute an adversarial perturbation. This is explained in detail in Section 3.1.1. To summarize, computing the true gradient of the loss needs access to the entire model, while an estimate of the gradient of the loss can be computed with just access to model probabilities.
>
> With regards to the practicality of our attacks and settings in which it could represent a threat, we emphasize that we demonstrate a real-world attack on Clarifai's Content Moderation and NSFW models. These are known to be deployed by Clarifai's clients. Using the access to class probabilities provided through a public API, we were able to create adversarial inputs with barely perceptible perturbations. An example is given in Figure 1 of the paper.
>
> To choose \delta, we performed a line search over a range of \delta values in order to estimate the gradient. Although a small value of \delta would give the best approximation, in reality using a very small value of \delta ends up with a bad approximation of the gradient, because the value of the cross-entropy loss does not change enough to be able to estimate the gradient at all. The logit loss is much more sensitive, and thus accurate estimates of it can be found using a smaller \delta.
>
> We clarify the threat model in Section 3 on Page 5 in the updated version. The only assumption we needed to perform a large majority of our attacks is access to the target model’s class probabilities. Only the PCA-based query reduction technique needs the extra assumption of access to a dataset representative of the training data.
>
> We do not make any assumptions on the training data of the public API, since we do not know what the model or data behind the API is. However, we were able to attack it in spite of this lack of knowledge.
>
> The models that use adversarial training were trained by us to evaluate the robustness of the proposed attack further. Even for our local models, we only assume black-box access, and we attack them without knowing the true gradient. For clarity’s sake, we have separated these two sets of experiments and clarified it in the updated version. The surrogate models used in the transferability attack are standard CNNs trained with the objective of minimizing the loss on the training set. This is the typical attack model assumed for transferability. In the case of both MNIST and CIFAR-10, the surrogate models used to transfer samples achieve classification accuracies close to that of the model which is attacked. The architecture and accuracy on benign data of all models are given in Appendix C.2.
>
> We do evaluate the attack success rate under a constraint on the maximum distortion possible. The maximum possible distortion is fixed by an L_{\infty} constraint on each pixel as is commonly done in the literature. However, most attacks don't perturb all the pixels, leading to a lower distortion than the maximum possible. Figure 4 in the Appendix shows some representative adversarial samples generated from our attacks with an L_{\infty} constraint of 0.3 for the MNIST data and 16 for the CIFAR-10 data. For comparisons with white-box attack given fixed distortion, we can compare the distortion for our attacks in Table 1 and that for white-box attacks in Table 7 in the Appendix. The distortion levels match almost exactly, thus the success rates are eminently comparable.
>
> The sensitivity of the attack success to \epsilon is shown in Figure 2 as well as in Figure 5 in the Appendix. The attack success increases as \epsilon increases. Figure 4 in the Appendix demonstrates that perturbation values of 0.3 for MNIST and 16 for CIFAR-10 do not cause significant difficulty in perception for humans, and can thus be safely chosen by an attacker. In the attack on Clarifai shown in Figure 1, we show that even an \epsilon value of 32 can be used safely by an attacker. We observe high attack success rates even at these perturbation limits.

---

> > ### Comment · AnonReviewer1 · 2018-01-08
> > **Good clarifications and revision**
> >
> > Thank you for your clarifications and changes. Most of my concerns were addressed. I appreciated the comparison to similar work [1] and the additional experiments to Chen at al. Overall this is an important problem, so I am happy to bump up my score.

---

### Official Review · AnonReviewer2 · 2017-11-28
**A very thorough empirical examination of practical black-box attacks on NN classifiers.**

**Rating:** 7
**Confidence:** 4

**Review:**

The authors consider new attacks for generating adversarial samples against neural networks. In particular, they are interested in approximating gradient-based white-box attacks such as FGSM in a black-box setting by estimating gradients from queries to the classifier. They assume that the attacker is able to query, for any example x, the vector of probabilities p(x) corresponding to each class.

Given such query access it’s trivial to estimate the gradients of p using finite differences. As a consequence one can implement FGSM using these estimates assuming cross-entropy loss, as well as a logit-based loss. They consider both iterative and single-step FGSM attacks in the targeted (i.e. the adversary’s goal is to switch the example’s label to a specific alternative label) and un-targeted settings (any mislabelling is a success). They compare themselves to transfer black-box attacks, where the adversary trains a proxy model and generates the adversarial sample by running a white-box attack on that model.  For a number of target classifiers on both MNIST and CIFAR-10, they show that these attacks outperform the transfer-based attacks, and are comparable to white-box attacks, while maintaining low distortion on the attack samples.

One drawback of estimating gradients using finite differences is that the number of queries required scales with the dimensionality of the examples, which can be prohibitive in the case of images. They therefore describe two practical approaches for query reduction — one based on random feature grouping, and the other on PCA (which requires access to training data). They once again demonstrate the effectiveness of these methods across a number of models and datasets, including models deploying adversarially trained defenses.

Finally, they demonstrate compelling real-world deployment against Clarifai classification models designed to flag “Not Safe for Work” content.

Overall, the paper provides a very thorough experimental examination of a practical black-box attack that can be deployed against real-world systems. While there are some similarities with Chen et al. with respect to utilizing finite-differences to estimate gradients, I believe the work is still valuable for its very thorough experimental verification, as well as the practicality of their methods. The authors may want to be more explicit about their claim in the Related Work section that the running time of their attack is “40x” less than that of Chen et al. While this is believable, there is no running time comparison in the body of the paper.

---

> ### Author Response · Authors · 2017-12-20
> **Quantitative evidence for running time comparison with Chen et al.**
>
> We thank the reviewer for the thoughtful comments. We have provided running times for our attacks in Appendix I.6 on Page 25. The same section also contains the results of Chen et al. in order to provide a quantitative comparison in the current updated version.

---

### Public Comment · (anonymous) · 2017-11-11
**Claim of novelty over Chen et al. (2017)**

The submission cites the paper by Chen et al. (2017), which proposes "ZOO: Zeroth Order Optimization based Black-box Attacks to Deep Neural Networks without Training Substitute Models". However, the submission goes on to claim the method of finite differences as a novel contribution, even though the cited paper by Chen et al. has already proposed it.

The "Gradient Estimation black-box attack based on the method of finite differences" presented in Section 3 and Section 3.1 of this submission, using a "two-sided approximation of the gradient", is identical to what is proposed in ZOO, which uses the "symmetric difference quotient to estimate the gradient" (Chen et al. 2017, equation 6).

---

> ### Author Response · Authors · 2017-11-14
> **Clarification of similarities and differences with regard to Chen et al. (2017)**
>
> The concurrent work from Chen et al. also proposes a black-box attack that uses queries from a model that exposes confidence scores. As you rightly note, both ZOO and our proposed methods do have in common that they use finite differences to estimate the derivative of a function. This shared part is a well-known method, for which we provide a citation (Spall, 2005).
>
> Beyond that, the attack methods proceed differently. We propose attacks that compute an adversarial perturbation, approximating FGSM and iterative FGS. On the other hand, ZOO approximates the Adam optimizer, while trying to perform coordinate descent on the loss function proposed by Carlini and Wagner (2016).
>
> We further provide new ways of reducing the number of queries required. Thus, our claim to novelty is not in using finite differences to estimate the gradient of a model, but the idea of estimating the gradient in a number of new query-reduced ways. Our work evaluates new attacks that use these estimates as well as known black-box attacks, as an additional contribution.
>
> Because of the relevance of Chen et al.’s work to the threat model, we will add a clarification in the “Related Work” section of the Introduction, as well as in Section 3, noting the fact that Chen et al. used the finite difference technique in a similar setting.
>
> Thank you for the comment!

---

> > ### Public Comment · (anonymous) · 2017-11-20
> > **Clarification comment**
> >
> > Hi, so it seems that the claimed novelty here is a way of reducing the number of queries using finite differences, however the ZOO attack also uses some novel techniques to reduce queries. It would be extremely useful to provide a side-by-side attack comparison with ZOO so we can infer which attack is more effective under various settings. You can find their code here https://github.com/huanzhang12/ZOO-Attack

---

### Author Response · Authors · 2017-12-20
**A revised version of the paper has been uploaded**

We have uploaded a revised version of the paper. In particular, the revised version contains the following changes:

1.  Shortened the Introduction by removing bulleted list of contributions
2. Clarified the practical relevance of situations where adversaries can have access to the loss of a model but not the gradient in the first paragraph of the Introduction
3. Provided a quantitative comparison with the running time of the closest related work by Chen et. al in Appendix I.6
4. Updated the anonymous link to more Clarifai model attack samples. It now directs to a zipped folder of attack images
5. Added Section 3.4 and Table 2 which provide both a quantitative and qualitative comparison of a range of gradient-free optimization methods to generate adversarial samples. In particular, we compare Gradient Estimation with Particle Swarm Optimization and Simultaneous Perturbation Stochastic Approximation
6. Description and results for the attacks on defenses and on the real-world Clarifai models are now in two separate sections for clarity

A number of other minor writing and presentation changes have also been made to improve the flow of the paper. We welcome further comments!

---

### Decision · Program_Chairs · 2018-01-29
**ICLR 2018 Conference Acceptance Decision**

**Decision:**

Reject

**Comment:**

The paper explores an increasingly important questions, especially showing the attack on existing APIs. The update to the paper has also improved it, but the paper is still not yet as impactful as it could be and needs much more comprehensive analysis to correctly appreciate its benefits and role.